behaviour, ecology

light pollution, insects, bioacoustics, circadian rhythm, biological clock

**Authors for correspondence:**
Amir Ayali
e-mail: ayali@post.tau.ac.il
Anat Barnea
e-mail: anatba@openu.ac.il

# Lifelong exposure to artificial light at night impacts stridulation and locomotion activity patterns in the cricket *Gryllus bimaculatus*

Keren Levy[1], Yoav Wegrzyn[1], Ronny Efronny[1], Anat Barnea[3] and Amir Ayali[1,2]

[1]School of Zoology, and [2]Sagol School of Neuroscience, Tel Aviv University, Tel-Aviv 6997801, Israel
[3]Department of Natural and Life Sciences, The Open University of Israel, Ra'anana 43107, Israel

KL, 0000-0002-0747-4992; AB, 0000-0002-1493-706X; AA, 0000-0001-6152-2927

Living organisms experience a worldwide continuous increase in artificial light at night (ALAN), negatively affecting their behaviour. The field cricket, an established model in physiology and behaviour, can provide insights into the effect of ALAN on insect behaviour. The stridulation and locomotion patterns of adult male crickets reared under different lifelong ALAN intensities were monitored simultaneously for five consecutive days in custom-made anechoic chambers. Daily activity periods and acrophases were compared between the experimental groups. Control crickets exhibited a robust rhythm, stridulating at night and demonstrating locomotor activity during the day. By contrast, ALAN affected both the relative level and timing of the crickets' nocturnal and diurnal activity. ALAN induced free-running patterns, manifested in significant changes in the median and variance of the activity periods, and even arrhythmic behaviour. The magnitude of disruption was light intensity dependent, revealing an increase in the difference between the activity periods calculated for stridulation and locomotion in the same individual. This finding may indicate the existence of two peripheral clocks. Our results demonstrate that ecologically relevant ALAN intensities affect crickets' behavioural patterns, and may lead to decoupling of locomotion and stridulation behaviours at the individual level, and to loss of synchronization at the population level.

## 1. Introduction

Artificial light at night (ALAN) is increasing worldwide by about 3–6% annually [1], with more than 80% of the world population living under light-polluted skies [2]. Awareness of the harmful effects of ALAN on living organisms is also increasing [3,4], including reports of changes in the length and quality of sleep [5,6], and in temporal activity shifts in mammals [5,7,8], birds [6,9,10], anurans [11] and marine species [12,13]. In insects, ALAN-induced changes in foraging activity could lead to higher predation risk [14,15], possible loss of camouflage [16–19] and increased mortality, partially owing to the attraction of many flying insects to light [20–22]. Insects also experience temporal disorientation under ALAN [17], leading to changes in community structure [1,14,16], and insect-induced environmental changes such as reduced pollination in meadows subjected to ALAN [23]. However, our knowledge of the effects of ALAN, specifically of its various intensities, on insect behaviour, temporal activity partitioning and fitness is far from complete.

Light is the most reliable *Zeitgeber*, synchronizing behavioural and physiological events through entrainment of the endogenous clock mechanism [24,25]. Diverse species, as well as different behaviours in the same organism, may react differently to the same light triggers or regimes, depending on their diurnal,

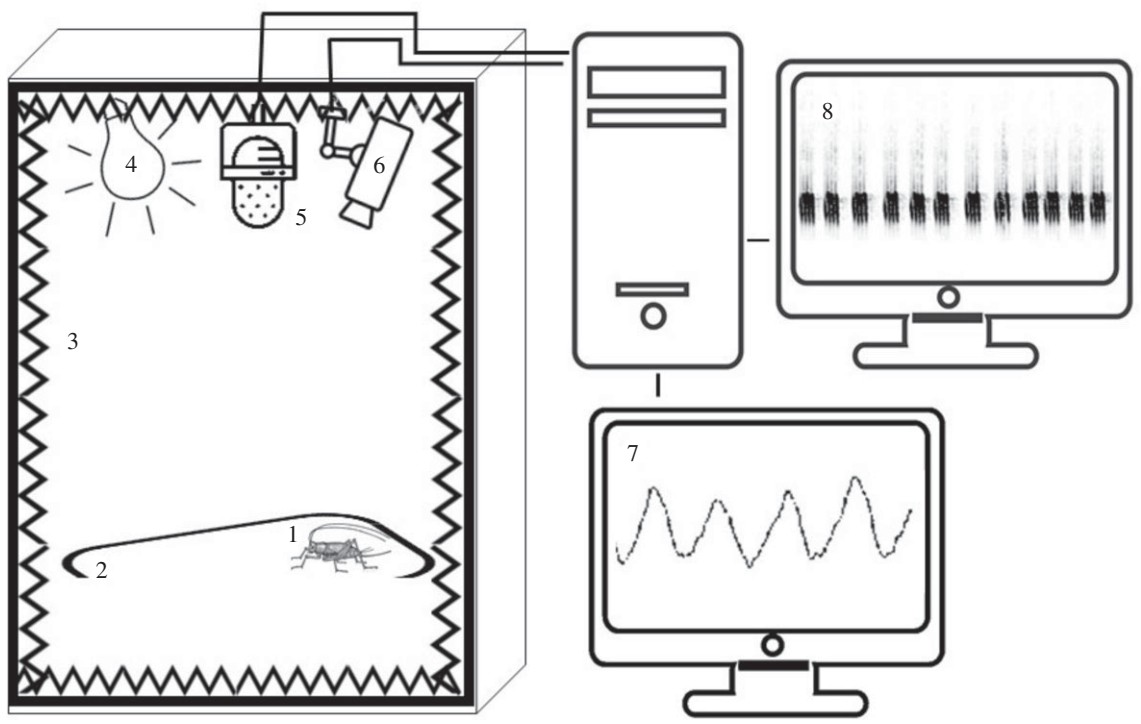

**Figure 1.** The experimental set-up: a cricket (1) was placed in a plastic box (2) in an anechoic chamber covered with acoustic insulation (3), equipped with a light source (4), a microphone (5) and an infrared camera (6). Sound and movement were recorded simultaneously and continuously (7, 8, respectively).

nocturnal or crepuscular way of life [26,27]. Circadian rhythms and daily activity patterns can be expressed in locomotion, sleeping, foraging, singing, gene expression and more [25,26,28]. Violation of the natural light–dark cycle may affect these rhythmical patterns. Changes in a light cycle or its disruption may induce either masking (an immediate response to the light stimulus that does not synchronize the pacemaker) [7,27] or entrainment (synchronization of the rhythms to environmental cues). Constant light conditions may evoke free-running (an endogenous rhythm with a period differing from 24 h that is not synchronized to any external cue), or arrhythmic behaviour [7,24,26,27].

Crickets (Gryllidae) have been widely used as models for the study of insect physiology, neurobiology and behaviour, including circadian activity [29,30]. They are known to demonstrate clear diel cycles in two fundamental behaviours, stridulation and locomotion [31–35]. In several cricket species, including *Gryllus bimaculatus*, temporal shifts in the insect's locomotor or stridulatory circadian activity have been reported following exposure to changes in illumination patterns [31–33]. These temporal shifts in activity patterns were in accord with changes in gene expression patterns connected to the circadian pathway [25,36,37]. Locomotion is important for the individual's fitness, especially in the context of foraging and the risk of predation. Stridulation, explicitly calling songs used for communication and female attraction, is crucial for the reproduction of the species. Any asynchrony in these behaviours among the population is critical. Nevertheless, the possible effects of ecologically relevant ALAN intensities on crickets, specifically the insects' susceptibility to different light intensities, have not been investigated to date.

Here, we studied the effects of exposing male *G. bimaculatus* crickets to lifelong ALAN on their stridulation and locomotion patterns. This is one of very few examples in which the two behaviours have been monitored simultaneously [32,34,35], conducing to a more comprehensive understanding of the ALAN-induced behavioural effects.

## 2. Material and methods

### (a) Rearing conditions

*Gryllus bimaculatus* crickets were reared under a constant temperature of $26 \pm 2°C$. Different experimental groups were reared from the egg stadium to hatching and through all life stages (i.e. lifelong) under one of four different light regimes. The rearing chambers were illuminated with white fluorescent light (CFL, NeptOn, 6500 K, 380–780 nm, peak: 547 and 612 nm; different light intensities were achieved by shading the light bulbs). All groups were exposed to 12 h daylight of 40 lux. Conditions between groups varied during the 12 h night period, as follows: (i) 12 h daylight : 12 h dark (LD, control), (ii) 12 h daylight : 12 h 2 lux ALAN ($LL_2$), (iii) 12 h daylight : 12 h 5 lux ALAN ($LL_5$), and (iv) 24 h constant daylight (LL). Crickets were fed three times a week with dog-food pellets and vegetables. The rearing boxes contained water flasks with absorbent cotton wool. Humidity was between 60 and 70%.

### (b) Experimental set-up

Male crickets, 3–7 days post-adult emergence, were individually assigned to one of four similar custom-made experimental anechoic chambers, enabling continuous and concomitant monitoring of the insect's locomotor and stridulation behaviours, and eliminating intraspecific communication (figure 1). The chambers were composed of Formica-laminated plywood plates (17 mm thick) covered with black acoustic foam (5 cm thick, density 33). Light was provided via a 5 W white CFL bulb (NeptOn, 6500 K, 380–780 nm, peak: 547 and 612 nm, see the electronic supplementary material, figure S1). Lighting conditions and the light regimes in the experimental chamber were similar to those that the insect had experienced in the rearing chambers (one of four, as described above). Particular care was taken to randomize the group-experimental chamber linkage. Light intensities were measured at four locations at the bottom of the chamber. Measurements were conducted at the cricket's eye level, at a distance of approximately 65 cm from the light bulb, using a digital light meter (TES-1337, TES, Taiwan). The light spectra were recorded using a Sekonic Spectromaster

C-700 (North White Plains, NY, USA). The mean ambient temperature was $25 \pm 2°C$. The anechoic chamber contained a plastic box with food, water and a paper cupcake mould, providing shelter and an elevated stridulation platform.

Acoustic recordings were conducted using a NT6 condenser microphone (Rhode Microphones, Sydney, Australia), placed 70–85 cm above the animal, connected to a M-track Quad amplifier (M-AUDIO, Cumberland, RI, USA), at 44 100 Hz and 16 bit depth, using a computer and RavenLite2.0.0 [38]. Locomotion activity was captured from above at 2 frames s$^{-1}$, by an infrared (IR) surveillance camera (constantly emitting IR light, peak: 799 nm), connected to a computer using the Active WebCam program (PY Software) for motion detection [39]. A threshold was defined as a change of more than 2% of pixels in the picture. IR enabled tracking the animals' locomotion under all light regimes, including scotophases.

## (c) Data processing and statistical analysis

Only recordings containing at least five consecutive days and nights of behavioural data were used. In cases in which the crickets did not perform either locomotion or stridulation behaviour in the first few days, or when system failure led to missing days of data, these days were removed from the analysis of that behaviour. Stridulation data extraction was conducted using 'R', version 3.4.1 [40], the 'Rraven' open source package [41] and RavenPro1.5 [42]. Data processing and statistical analyses were conducted in Python v. 3.7 (PyCharm, JetBrains), MATLAB (The MathWorks, MA, USA), SPSS v. 21 (IBM Corp., Armonk, NY, USA) and Prism 8 (GraphPad Software, San Diego, CA, USA). The number of stridulation syllables and locomotion events were assessed per animal in 10 min bouts. For diurnal and nocturnal quantitative activity comparisons, values were normalized for each individual by dividing that individual's values by its own mean value. For behaviour, rhythmicity, and period analyses and comparisons, values were normalized for each individual by dividing that individual's values by its own maximum value, resulting in an activity index ranging from 0 (no activity) to 1 (maximal activity). Periodogram analyses of the activity rhythm periods were determined using the ImageJ plugin ActogramJ [43].

Comparisons of nocturnal and diurnal activity, as well as the absolute difference within groups, were conducted with the Kruskal–Wallis test, followed using a Dunn's test. Control nocturnal and diurnal activities of the same individuals were compared by a paired $t$-test. The analysis of the nocturnal and diurnal activity in the constant daylight (LL) crickets was based on the objective day and night periods. The analyses of the medians and variance of the cycle periods were conducted using MATLAB. Values over three times the absolute deviations from the scaled median were considered as outliers and removed. The median and variance of the periods were compared using the Kruskal–Wallis test and the Brown–Forsythe test for equality of variance, respectively. Both tests were followed by a Bonferroni multiple comparison *post hoc* test. Spearman's rank-order correlation was used to evaluate the relationship between stridulation and locomotion behaviour. Differences between both behaviours were assessed using the Kruskal–Wallis test. A $\chi^2$ test was used to evaluate a possible connection between ALAN intensity and rhythm types. It should be noted that most of our data are characterized by inequality of variance and/or by non-normal distribution. Hence, medians rather than means were often presented, and non-parametric statistical tests were used.

The mean acrophase (the time at which the peak of a rhythm occurs) of 5 days was calculated for each animal, period and behaviour using the CosinorPy package [44]. Circular statistical analyses were conducted using the Oriana software, v. 4 (Kovach Computing Wales, UK) [45]. Phases were averaged per treatment and behaviour. The Mardia–Watson–Wheeler test was used for distribution comparisons among treatments and the Hotellings paired test assessed phase differences within the same individual. Both tests were followed by a Bonferroni correction.

## 3. Results

### (a) Crickets' morphological characteristics

The lifelong lighting conditions had a somewhat inconsistent effect on the crickets' morphological characteristics (size, weight and morphometric relationships; electronic supplementary material, figure S2). While we observed a significant effect on the body condition index (Kruskal–Wallis, $p < 0.038$), no significant differences were found among the different treatments (Dunn's *post hoc* multiple comparison, $p > 0.9$ for all comparisons; electronic supplementary material, figure S2).

### (b) Temporal patterns of stridulation and locomotion behaviours

The control crickets exhibited an activity rhythm of 24 h, with stridulation behaviour displayed predominantly at night (figure 2$a$(i)(ii); 5.6% diurnally and 94.4% nocturnally, paired $t$-test, $n = 15$, $p < 0.0001$), and locomotor activity mainly during the day (figure 2$b$(i)(ii); 68.3% diurnally and 31.7% nocturnally; paired $t$-test, $n = 11$, $p < 0.05$, electronic supplementary material, tables S1 and S2). Figure 2$a$(iii),$b$(iii) presents examples of a double-plotted actogram of two individuals representing typical LD behaviours.

A quantitative comparison of diurnal and nocturnal activity in the different experimental groups revealed that the median of the normalized diurnal, as well as nocturnal LD stridulation activity level significantly differed from the LL$_5$ and LL treatments ($p < 0.007$ for all, Kruskal–Wallis test with Dunn's multiple comparisons; figure 3$a$(i); electronic supplementary material, table S1). Significant differences in diurnal locomotor activity level were found between LL and all other treatments, while for the nocturnal locomotor activity levels, differences were significant only between LL and both LL$_2$ and LL$_5$ ($p < 0.01$ for diurnal locomotion, $p < 0.02$ for nocturnal locomotion, Kruskal–Wallis test with Dunn's multiple comparisons, figure 3$b$(i); electronic supplementary material, table S1). In both behaviours, the effects of the light regime and ALAN were also apparent when comparing the difference between diurnal and nocturnal activity levels (Kruskal–Wallis test, $p < 0.0001$, figure 3$a$(ii),$b$(ii); electronic supplementary material, table S1).

The different experimental groups also differed in the temporal patterns of the two monitored behaviours, with a clear ALAN-dependent decrease in rhythmicity (figure 3$a$(iii), $b$(iii)). This was observed in both stridulation (figure 3$a$(iii)) and locomotion (figure 3$b$(iii)) behaviours. For both behaviours, averaging the free-running data of the different LL individuals resulted in a practically absent rhythm.

### (c) Period and group variance significantly differed among treatments

The data in figure 4 present the analysis of the periods of the recorded behavioural patterns. Only individuals that exhibited a significant period (electronic supplementary material, table S1) were included in this analysis. The median period of stridulation activity cycles differed significantly between the LD crickets and both the LL$_5$ and LL groups (24.0 h, 25.17 h and

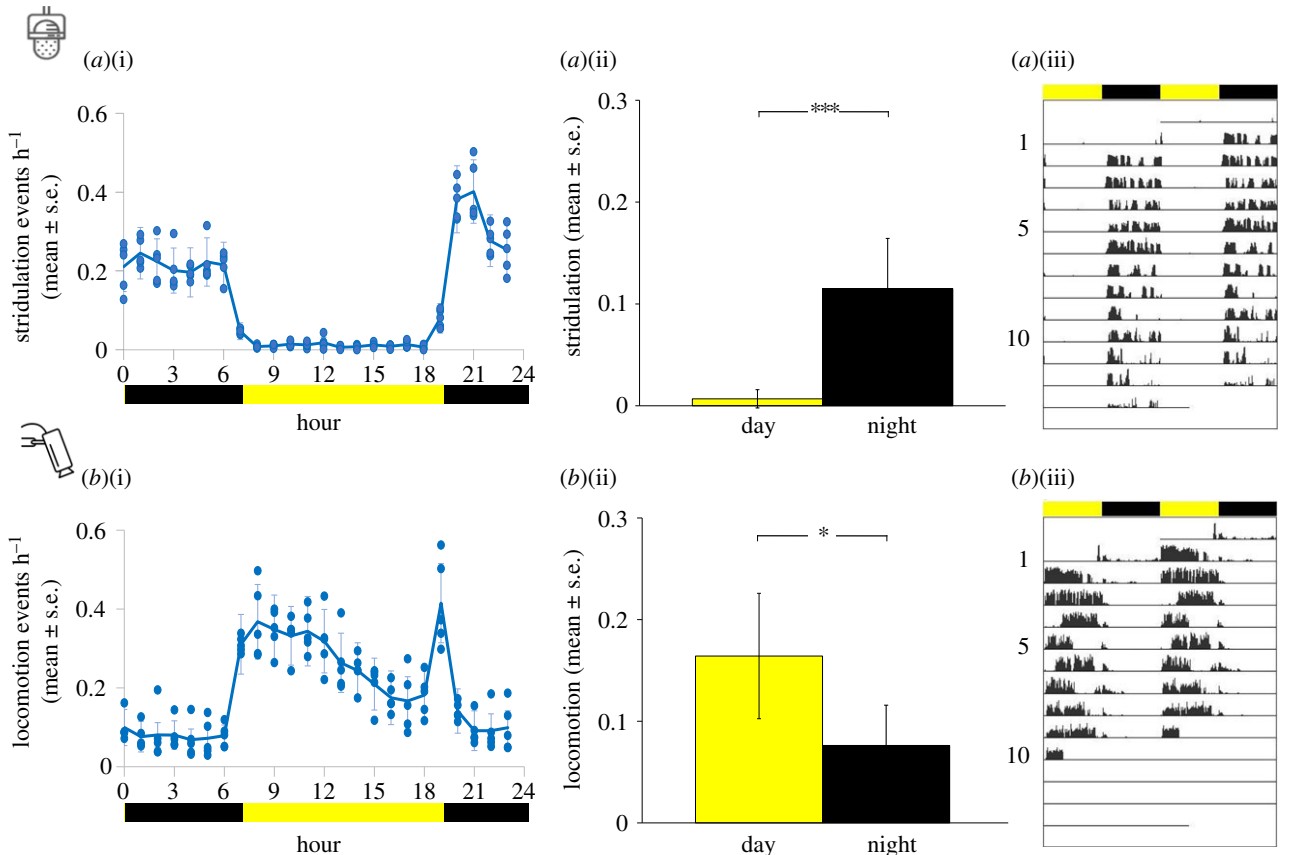

**Figure 2.** Light-dependent behaviour of control animals. Normalized activity cycle of stridulation (*a*)(i) and locomotion (*b*)(i) (mean ± s.e.). Normalized day and night activity levels (*a*(ii),*b*(ii); mean ± s.e.). Stridulation; *n* = 15; locomotion; *n* = 11; *$p < 0.05$, ***$p < 0.001$. Values were normalized by dividing each individual's values by its own maximum value, resulting in an activity index ranging from 0 (no activity) to 1 (maximal activity). Double-plotted actograms representing typical LD behaviours of two individuals (*a*(iii),*b*(iii)). Yellow and black bars indicate diurnal and nocturnal phases, respectively. (Online version in colour.)

25.67 h, respectively; *n* = 15, 19 and 17, respectively; Bonferroni correction for multiple comparisons). The LL group also differed from $LL_2$ (24.67 h; *n* = 19). Furthermore, the variance in the period of stridulation of all three ALAN treatments significantly differed from that in the LD treatment (figure 4*a*; Brown–Forsythe test with the Bonferroni correction).

The median period of locomotor activity differed significantly between LL and all three other treatments (figure 4*b*; 25.5 h in LL; 24.0 h in LD, $LL_2$, and $LL_5$; *n* = 24, 11, 25 and 13, respectively; Bonferroni correction). In addition, the variance in the period of locomotor behaviour differed significantly between LL and both LD and $LL_2$ treatments (figure 4*b*; Brown–Forsythe test with the Bonferroni correction).

No correlation was found between stridulation and locomotion activity cycle periods (Spearman's rank-order correlation, $r_{s47} = 0.204$, $p = 0.160$). A similar lack of correlation persisted when examining the different treatments separately. Moreover, the calculated individual absolute difference between the rhythm periods of stridulation and locomotion was found to differ significantly among treatments (Kruskal–Wallis test, $\chi^2_{3,49} = 9.75$, $p = 0.021$), suggesting the asynchrony of these two periods.

### (d) Artificial light at night elicited three types of activity patterns

Three types of activity pattern were observed in the different treatments, in both stridulation and locomotion activity: synchronized rhythms (periods of 24 h; figure 5*a*(i)), free-run rhythms (periods different from 24 h; figure 5*a*(ii)) and

arrhythmic activity (lack of any period; figure 5*a*(iii)). The proportions of these types of rhythms in each of the experimental groups reflect a clear effect of the ALAN intensities on the behavioural activity (figure 5*b*(i)(ii); electronic supplementary material, table S2). Both the percentage of synchronized stridulation and locomotion rhythms dropped steeply from the LD to all ALAN treatments (figure 5*b*(i)(ii); electronic supplementary material, tables S1 and S2). An opposite trend was observed for the free-run rhythms, which were lowest in the control and high in all other treatments. In both behaviours, arrhythmic activity was not observed in LD, while common in LL, and also present in $LL_2$ and $LL_5$ (figure 5*b*(i)(ii); electronic supplementary material, table S2). The occurrence of an activity type was not associated with the type of behaviour (i.e. stridulation or locomotion; $\chi^2_{2,190} = 0.00$, $p = 1$). However, the activity type was significantly dependent on the treatment (i.e. the light regime ($\chi^2_{6,190} = 105.57$, $p < 0.0001$).

### (e) Phase distributions significantly differed among treatments

Only individuals that exhibited a significant period (electronic supplementary material, table S1) were included in the acrophase analysis of stridulation and locomotion behaviour (figure 4*c*,*d*; electronic supplementary material, table S3). In stridulation behaviour, vector length was gradually reduced with ALAN intensity (LD: 0.81, $LL_2$: 0.40, $LL_5$: 0.42 and LL: 0.30), while in locomotion behaviour, the intensity was reduced with $LL_5$ and LL only (LD: 0.67, $LL_2$: 0.75, $LL_5$: 0.47, and LL: 0.26; electronic supplementary material,

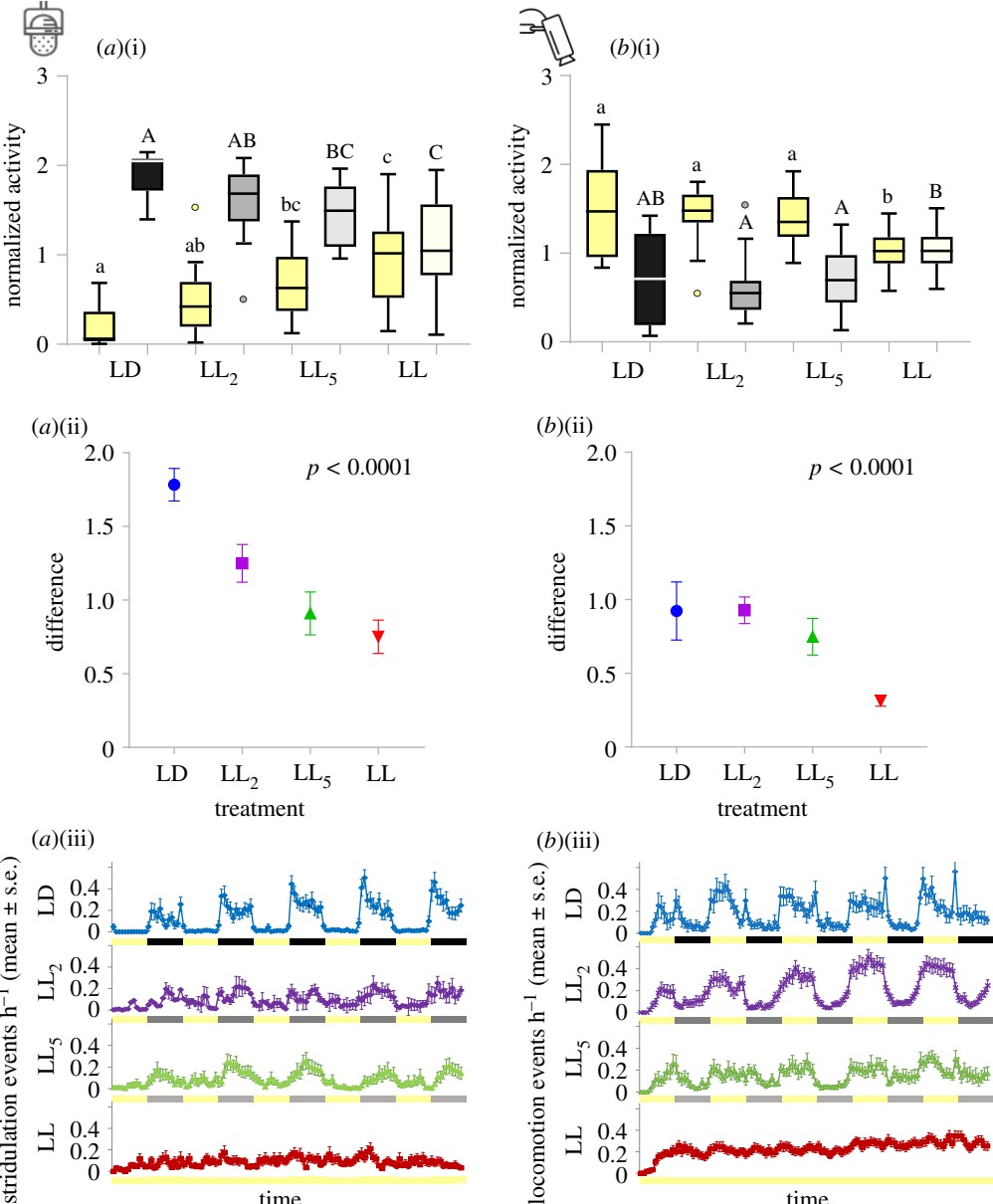

**Figure 3.** ALAN intensities induce variation in the temporal activity and rhythmicity of crickets. Quantitative comparisons of diurnal (yellow, small letters) and nocturnal (black and grey, capital letters) stridulation ($a$)(i) and locomotion ($b$)(i) activity under LD ($n = 15$, $n = 11$, respectively), LL$_2$ ($n = 20$, $n = 25$, respectively), LL$_5$ ($n = 20$, $n = 17$, respectively) and LL ($n = 21$, $n = 40$, respectively) treatments (mean ± s.e.). Different letters represent statistical differences. The differences between nocturnal and diurnal activity of stridulation ($a$)(ii) and locomotion ($b$)(ii) under LD (blue), LL$_2$ (purple), LL$_5$ (green) and LL (red) treatments (mean ± s.e.). Normalized activity cycles per hour of ($a$)(iii) stridulation and ($b$)(iii) locomotion under LD (blue), LL$_2$ (purple), LL$_5$ (green) and LL (red) treatments (mean ± s.e.). Five consecutive days (yellow) and nights (black, grey) are presented (LD show the same data as in figure 2$a$(i) and $b$(i)). Values for $a$(i)(ii) and $b$(i)(ii) were normalized by dividing each individual's values by its own mean value, while values for $a$(iii) and $b$(iii) were normalized by dividing each individual's values by its own maximum value, resulting in an activity index ranging from 0 (no activity) to 1 (maximal activity). (Online version in colour.)

table S3). In addition, the variance in both activities became larger with higher ALAN intensity (figure 4$c$,$d$; electronic supplementary material, table S3), while the mean phase vector angles remained steady in the first three treatments (LD, LL$_2$, LL$_5$), indicating an overall loss of phase synchronization with higher ALAN intensity. The mean phase vector angles of stridulation activity differed significantly between the LL and all other treatments and that of the locomotion activity differed between LL, LD and LL$_2$ (Watson–Williams $F$-test, $p < 0.0167$). However, an ALAN-intensity-dependent phase distribution was observed, with LD narrowly distributed and LL evenly distributed. Stridulation LD differed significantly from that of LL$_2$ and LL (Mardia–Watson–Wheeler test, $p = 0.003$ and $p = 0.001$, respectively), while in

locomotion behaviour, the LL treatment differed significantly from that of the LL$_2$ and somewhat also from LD (Mardia–Watson–Wheeler test, $p < 0.0002$ and $p = 0.033$, respectively). A moderate circular–linear correlation was found between the treatments and both activity behaviours ($r = 0.4$, and $r = 0.44$, respectively).

Conducting the same analysis only on animals for which we had data for both stridulation and locomotion (for $n$, see the electronic supplementary material, table S1) enabled the comparison of the phase between the two behaviours. This revealed yet another effect of ALAN: in both, LL$_5$ and LL, the significant difference between the stridulation and locomotion phases, observed under control conditions, was lost (Hotelling's paired test, LD$_{stridulation}$ − LD$_{locomotion}$: $p <$

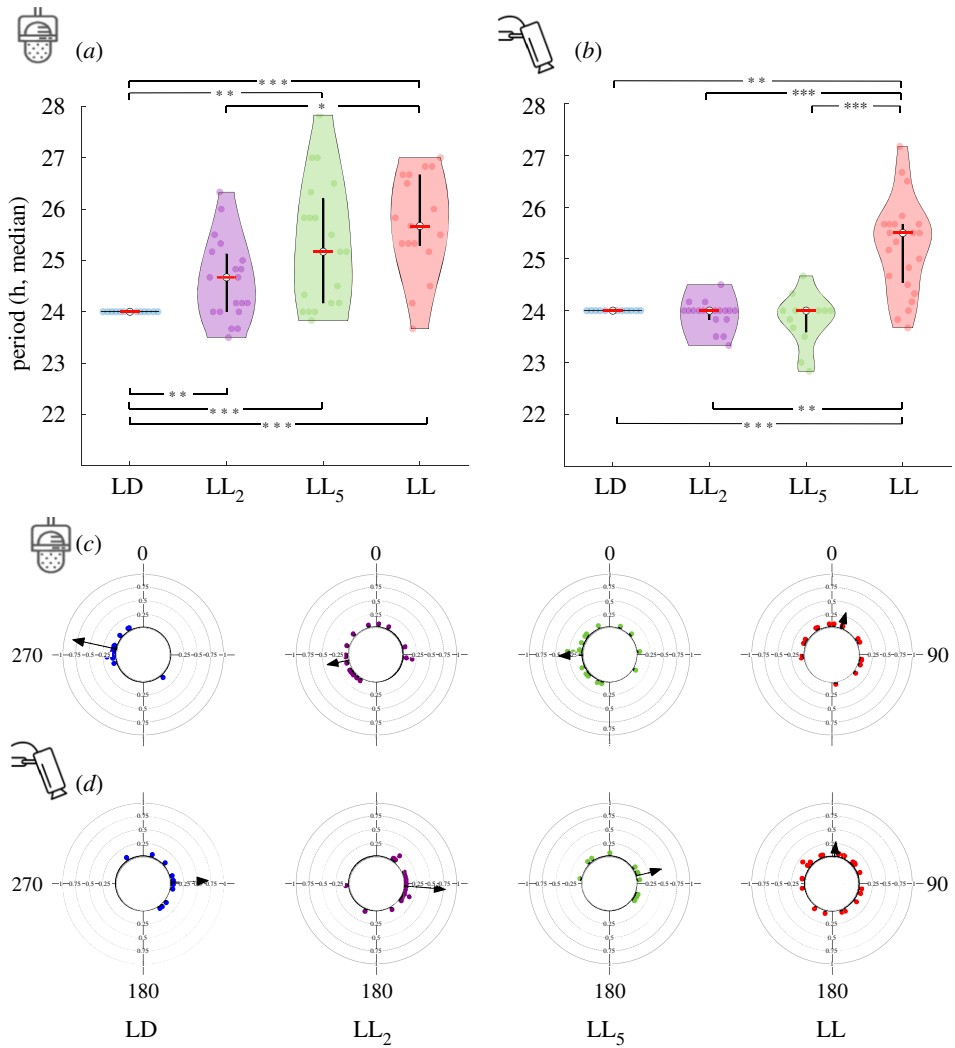

**Figure 4.** ALAN affects the individual periods, the medians and variance of stridulation (*a*) and locomotor (*b*) activity of the groups. Red lines represent medians, and individual males are plotted in coloured dots. Coloured areas are kernel histogram, rotated by 90°. Black vertical lines represent the inter-quartile range 25% below and 25% above the median. Upper asterisks: differences between medians; bottom asterisks: differences between variances. *$p < 0.05$, **$p < 0.01$, ***$p < 0.001$. ALAN affects the acrophase of individuals' behaviour, and the group variance in stridulation (*c*) and locomotion (*d*) behaviour. Each point in the circular plots represents the mean phase of 5 days of activity of an individual male cricket. Grid lines = 0.25. The black arrow represents the mean vector of phase, calculated for the entire experimental group. (Online version in colour.)

0.0001, $LL_{2stridulation} - LL_{2locomotion}$: $p < 0.0001$, $LL_{5stridulation} -$ $LL_{5locomotion}$: $p = 0.136$, $LL_{stridulation} - LL_{locomotion}$: $p = 0.765$).

## 4. Discussion

### (a) Crickets as a model for artificial light at night research

Despite the use of crickets as model insects to study behaviour, physiology and neurobiology for over a century [30,46], simultaneous and parallel long-term monitoring of individual insect stridulation and locomotion has, to our knowledge, never been carried out in this insect. Our unique custom-made set-up enabled such monitoring under various light conditions, providing an excellent signal-to-noise ratio. Our findings add to the studies of Tomioka & Matsumoto [25] in supporting *G. bimaculatus* as a suitable model insect for ALAN research.

In the present study, we report on the temporal partitioning of field cricket behaviour, with stridulation being

predominantly nocturnal and locomotion predominantly diurnal. This is in contrast with previous studies in the same species that demonstrated the diurnal locomotion of nymphs, but nocturnal locomotor activity of adults [47]. The growth chamber and experimental set-up temperatures were 24–28°C. Hence, the above discrepancy cannot be explained by rhythm reversal as a result of exposure to low temperatures [48]. The natural behaviour of the species might differ among different laboratory colonies, and reflect certain adaptive colony-specific behaviours. Additionally, there may be some indirect effects of the different illumination and motion detection methods used in the different laboratories. While, as noted, our method revealed mostly diurnal locomotion, we did observe in the LD group a peak in locomotion at sunset, preceding the beginning of stridulation behaviour, which is in accord with the actograms shown in [47, fig. 1 therein]. Although this has little impact on the main findings reported here, this issue deserves further investigation.

When studying the effects of light on behaviour, the visual system of the model insect should be taken into consideration. The following spectral types have been described in the cricket

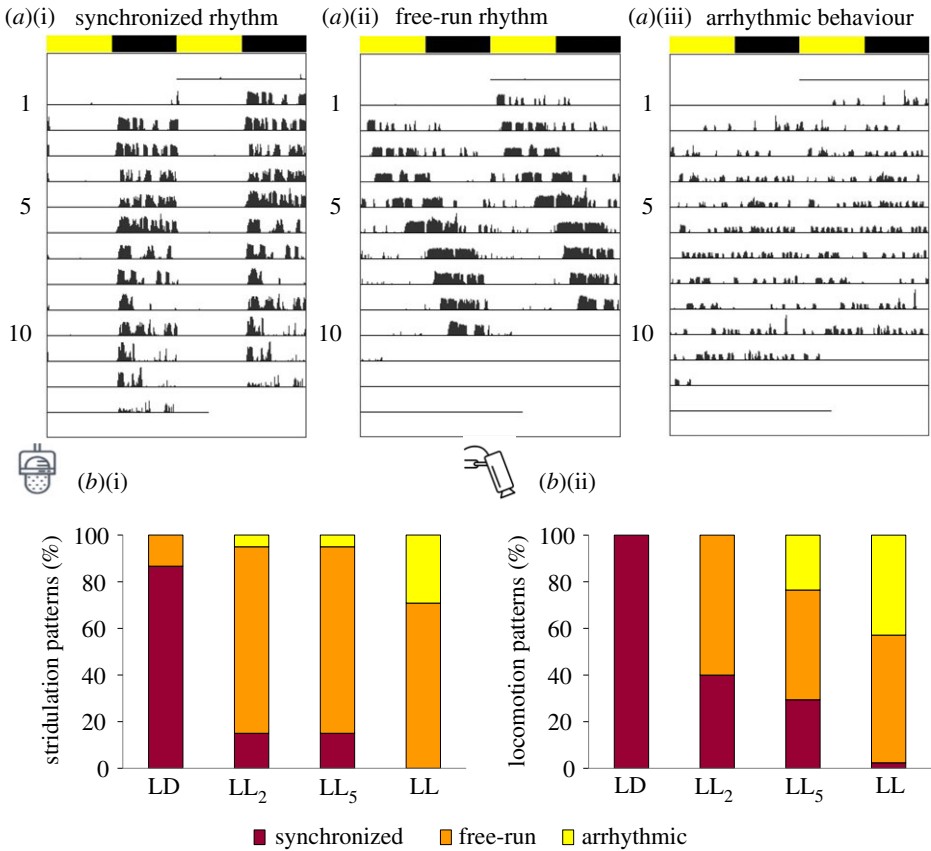

**Figure 5.** ALAN induces different types or activity patterns. Double-plotted actograms demonstrating activity patterns of adult male crickets: (*a*)(i) a synchronized rhythm of 24 h; (*a*)(ii) a free-run rhythm of 26 h; and (*a*)(iii) arrhythmic behaviour. Yellow and black bars indicate diurnal and nocturnal phases, respectively. Percentage of synchronized (dark red), free-run (orange) and arrhythmic (yellow) stridulation (*b*)(i) and locomotion (*b*)(ii) behaviour in male adult crickets exposed to the different lifelong ALAN intensities. (Online version in colour.)

*G. bimaculatus*: UV (peak: 332 nm), blue (peak: 445 nm) and green (peak: 515 nm), resulting in a spectral sensitivity of up to 600 nm [49], with the performance of the photoreceptors leading to the conclusion that the crickets' eye has evolved for signal processing in dim light [50]. In our experiments, shading the light bulbs in order to create lower light intensities resulted in a relative decrease in the blue wavelength (400–500 nm) intensity and an increase in the red wavelength (greater than 700 nm) intensity. Interestingly, while this increase in red wavelength had no influence on the experiments, as crickets do not possess the ability to see red light, a clear effect of $LL_2$ was observed despite the decrease in the blue wavelength.

As reported, the different lifelong lighting regimes used in our study did not reveal any clear effect on the general properties of the crickets' morphology. This aspect should also be further explored; see available data showing larger femur length in ALAN-exposed *Teleogryllus commodus*, in [51]).

## (b) Ecological perspectives

Despite increasing concern regarding the harmful effects of ALAN on humans and other animals [18,52–54], there is still insufficient knowledge regarding its impact on insects, which constitute a major bio-indicator of environmental changes and pollution [55,56]. The ALAN intensities investigated here had been previously reported as prevalent and ecologically relevant in urban environments, affecting the behaviour and ecology of various species [5,9,17,54,57], as well as having a 'sink' effect on flying insects attracted

towards street lights [3,18,20,58,59]. We explored the effect of ALAN on two key behaviours: locomotion, which is important in foraging; and stridulation, which is a major component of the cricket's courtship behaviour, sexual selection and intraspecific communication [60–66]. The timing of these behaviours, as well as their synchrony among individuals, has an important impact on both the fitness of the individual and the reproductive success of the population.

In our study, both behaviours presented a robust rhythm in LD, with clear diurnal and nocturnal partitioning, while exposure to lifelong ALAN resulted in free-run behaviour or even arrhythmicity, and consequent loss of quantitative and temporal partitioning and higher group variance. Initially, the acrophase of both behaviours appears to reveal some resilience, despite the changing periods. However, a deeper examination revealed a trend of ALAN-induced shortened mean vectors with growing phase variance, and even loss of phase synchrony among stridulation and locomotion activity in the same individual. Interestingly, our findings suggest differences between the two studied behavioural patterns, specifically in their susceptibility to ALAN. Stridulation behaviour was strongly affected even by low ALAN intensity, while the period of the locomotion activity patterns revealed some resilience to ALAN exposure, demonstrating a gradual ALAN-dependent change. It should be noted that a period of 24 h does not necessarily reflect a stable endogenous cycle but could also be the result of masking. Under natural conditions, however, other factors, such as the diurnal temperature, as well as conspecific activity, might contribute to population synchronization [67]. We, therefore,

conclude that *G. bimaculatus* populations are vulnerable to low levels of ALAN, raising concerns about the possible detrimental ecological effects of light pollution on this as well as other insects.

## (c) Perspectives for future research

Some of our findings indicate major routes for future research into circadian rhythms and their underlying mechanisms in the cricket. For example, while the period of stridulation and locomotion has been shown to correlate in other species [32,35], no correlation of either the period or the phase was found in our present study. The differential effects of lifelong exposure to increasing ALAN intensities on the activity cycles and phases of these two behavioural patterns in a single individual may indicate the decoupling of possibly two peripheral clocks.

Our experiments were conducted under laboratory conditions, with a relatively low daylight intensity in order to avoid both high chamber temperatures and visual discomfort to the insects (which live a cryptic life). The chosen light intensity may not faithfully represent natural light conditions. Furthermore, the experimental illuminations were turned on and off abruptly, lacking the natural gradual processes of sunrise and sunset. It is possible that the free-run pattern was triggered not only by the nocturnal light intensity *per se*, but also by the differences in intensity between the diurnal and nocturnal light. Hence, it is important to conduct such experiments also under natural conditions, in order to determine whether and how these ALAN intensities might affect the field crickets in their natural habitat.

Data accessibility. Electronic supplementary material and data are available from the Dryad Digital Repository: https://doi.org/10.5061/dryad.wm37pvmmf [68].

Authors' contributions. K.L.: conceptualization, formal analysis, investigation, methodology, validation, writing—original draft, writing—review and editing; Y.W.: software, validation, visualization; R.E.: software, validation, visualization; A.B.: conceptualization, funding acquisition, supervision, writing—original draft, writing—review and editing; A.A.: conceptualization, funding acquisition, supervision, writing—original draft, writing—review and editing. All authors gave final approval for publication and agreed to be held accountable for the work performed therein.

Competing interests. The authors have no competing interests related to this study.

Funding. K.L. is grateful to the Cornell Bioacoustics Research Program for supporting attendance to the 2018 Sierra Nevada Sound Recording and Analysis Workshop. This research was funded by The Open University of Israel Research Fund.

Acknowledgements. We thank Izhak Idelstein, Stan Moaraf, Yael Ballon, Daniel Knebel and the staff of the I. Meier Segals Garden for Zoological Research at Tel Aviv University for their assistance, and Eran Tauber for his valuable comments on the manuscript.

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
