## [Peer Review File · Proceedings of the Royal Society B: Biological Sciences]

Review History

RSPB-2021-0233.R0 (Original submission)

Review form: Reviewer 1

Recommendation

Reject – article is scientifically unsound

Scientific importance: Is the manuscript an original and important contribution to its field?

Good

General interest: Is the paper of sufficient general interest?

Good

Quality of the paper: Is the overall quality of the paper suitable?

Poor

Is the length of the paper justified?

Yes

Should the paper be seen by a specialist statistical reviewer?

No

Do you have any concerns about statistical analyses in this paper? If so, please specify them explicitly in your report.

Yes

It is a condition of publication that authors make their supporting data, code and materials available - either as supplementary material or hosted in an external repository. Please rate, if applicable, the supporting data on the following criteria.

Is it accessible?

Yes

Is it clear?

Yes

Is it adequate?

Yes

Do you have any ethical concerns with this paper?

No

Comments to the Author

Reviewer comments

Title: Lifelong exposure to artificial light at night impacts stridulation and locomotion activity patterns in the cricket

This study monitored behavior of crickets under different intensities of ALAN. I believe it should make a good contribution to the field of circadian and ALAN research. However, I have concerns about the lack of clarity of the methods and analyses that are outlined in more detail below.

Briefly, sample numbers, number of chambers (both rearing and experimental), etc are unclear in methods. Although light was applied at the adult stage, it is not a lifelong application if not for the larval stage and behavior monitoring only occurred during 5 days, therefore, I would be cautious about the use of "lifelong" in all places of the manuscript. Additionally, the analyses is comparing the same individual in paired T-tests, whereas different individuals were exposed to different treatments. In order to tell if light had an effect, treatment needs to be compared (see below). The circadian periodicity analyses also should be updated with circadian analyses rather than just mean and variance, that way you can specify shift, amplitude, etc.

Abstract: unclear why abstract was divided into different paragraphs

L25: explain these differences more clearly; same as in L27: the descriptions of differences in abstract is a bit vague

L37: is there a global statistic? Or is the cricket endemic to Europe?

L47: could you specify what exactly is missing?

Methods:

L77: were daylight and nightlight the same? All fluorescent?

L80: what does LA stand for? Why not use ALAN? Does the A mean less light? Could abbreviate to LLdim2lux?

In general through ms and also in title: it states that crickets were under lifelong exposure but it doesn't say at which point in their life cycle light treatment was applied. Also during the larval stage? At some point it says males 3-7 days post adult emergence, but that would mean they would still have at least 2 weeks to live but monitoring was only done for 5 days?

How many chambers were there? Did the crickets rotate in the chambers?

L89: says that 5W bulb was used in chambers, and that it was the same light as rearing light. But unclear earlier that type of rearing light it was.

How many rearing chambers were there? Were individual crickets reared in these chambers or grouped? It might be problematic if they were in groups and exposed to light as treatment is then by chamber and not by individual.

Either way, there needs to be a control (random effect) of rearing chamber (if more than one).

How many male crickets were used in total? How many were actually analyzed? First time sample numbers were evident was in L133. N=11 vs 15?

Why were crickets analyzed using paired T-tests with themselves? Shouldn't nocturnal activity be analyzed by treatment to see if treatment had an effect and used with rearing conditions as a random effect? Here you can only see if diurnal and nocturnal activity differed, and of course they did, in BOTH groups. To see if control and treatment conditions differed in nocturnal activity, you need a linear model with treatment as a fixed effect.

In figure 1, also unclear how data were normalized.

Based on FigS1, it's not just that the intensities that changed but also wavelengths, 2lux light seems to have more red wavelengths than other treatments. This should be discussed in the manuscript.

FS2 L141: were before weights and lengths measured? Can you do a body condition index based on single weight and length metrics rather than all of them?

L147: I don't understand how there is such a difference in sample numbers across different analyses, some had n=19, but then n=15, n=11. Could authors please have a table of sample numbers somewhere and explain the discrepancies?

L119: period analyses is more than just mean and variance but also cycles. Highly recommend using circadian analyses to look at circadian behavior. For example, Cosinor v.1.1 package in R (Barnett and Dobson, 2010) can be used to test treatment effects on amplitude (i.e. difference between peak and mean value of wave) and acrophase (i.e. time of peak expression in wave). You can also use Rhythmicity Analysis Incorporating Nonparametric (RAIN) and Detection of Differential Rhythmicity (DODR) analyses to test for differences between the circadian rhythmicity and peak phase of control and ALAN groups (Thaben and Westermarck, 2016, 2014).

Fig4: it would be nice to know which ALAN treatment resulted in which behaviors.

L200: I feel that some of the explanations of the importance of these behaviors should be in the intro rather than the discussion

Review form: Reviewer 2 (Roy H. A. van Grunsven)

Recommendation

Major revision is needed (please make suggestions in comments)

Scientific importance: Is the manuscript an original and important contribution to its field?

Good

General interest: Is the paper of sufficient general interest?

Good

Quality of the paper: Is the overall quality of the paper suitable?

Acceptable

Is the length of the paper justified?

Yes

Should the paper be seen by a specialist statistical reviewer?

No

Do you have any concerns about statistical analyses in this paper? If so, please specify them explicitly in your report.

No

It is a condition of publication that authors make their supporting data, code and materials available - either as supplementary material or hosted in an external repository. Please rate, if applicable, the supporting data on the following criteria.

Is it accessible?

Yes

Is it clear?

No

Is it adequate?

No

Do you have any ethical concerns with this paper?

No

Comments to the Author

This is a very nice paper that shows an impact of ALAN that was previously unknown. ALAN clearly affects the timing of the behavior of crickets and the impact increases with intensity. Especially that the two behaviors studied seem to respond independently is important as this indicates that ALAN can result in desynchronization of different behaviors. The results are very convincing in that perspective.

One aspect that deserves more attention is the choice of light levels. It is shortly discussed that the diurnal light level is "relatively low" with 40 lux. A typical summer day is 2000 (very cloudy) to >100.000 lux. A value of 40 lux is overcast at dusk. I understand the practical objections to high light levels however and this is made explicit. The 2 and 5 lux levels are reasonable. These are levels you find near illumination and in urban areas. It would be interesting to know the impact of lower (sky-glow) levels but that is a different question. However, the control treatment is 0 lux. This raises a few questions. The first is how do you monitor movement without any light source? Or was there an IR-source? More importantly 0 lux is far from a natural control. These crickets are active at night and use the low levels of natural nocturnal light to see. Having true darkness deprives them of sight. So the control that should reflect the natural condition, high light level (>1000 lux) during the day and low light levels (0.0001-0.05 lux) at night, is low light levels during the day and unnatural darkness at night. This might very well affect the activity of the crickets.

The presentation of the light treatment in the materials and methods is also very limited. If only one type of light source was used how were the different light levels achieved? How did you measure these, where (middle, or multiple spots) and with which device? It is written that the CFL bulb emits 40 lux, the appropriate unit for luminous flux is lumen, not lux. Or did you measure illumination at the bottom of the cage? In that case it is only a part of what the bulb emits but in lumen (and more relevant).

The blue part of the spectra differs quite substantially between the light levels, this might be relevant for entrainment and warrants discussion. I would also suggest giving the spectral measurements in Dryad, this allows others to calculate different aspects of your treatment (photon flux, intensity based on the spectral sensitivity of the eyes of crickets etc.), a graph gives insight but does not allow for further analysis. It would also be helpful to upload the raw data with labels that are easily interpretable.

Review form: Reviewer 3

Recommendation

Major revision is needed (please make suggestions in comments)

Scientific importance: Is the manuscript an original and important contribution to its field?

Good

General interest: Is the paper of sufficient general interest?

Good

Quality of the paper: Is the overall quality of the paper suitable?

Good

Is the length of the paper justified?

Yes

Should the paper be seen by a specialist statistical reviewer?

No

Do you have any concerns about statistical analyses in this paper? If so, please specify them explicitly in your report.

No

It is a condition of publication that authors make their supporting data, code and materials available - either as supplementary material or hosted in an external repository. Please rate, if applicable, the supporting data on the following criteria.

Is it accessible?

Yes

Is it clear?

Yes

Is it adequate?

Yes

Do you have any ethical concerns with this paper?

No

Comments to the Author

This is an interesting paper dealing with the effect of artificial light exposure at night (ALAN) on daily rhythms of stridulatory and locomotor activities. The authors showed that ALAN had significant and differential effects on the two rhythms, suggesting that ALAN affects the behavioral rhythms in this species and that the two rhythms were controlled by separate mechanisms. The experiments were appropriately performed, the results are interesting and discussed reasonably. However, the manuscript needs some improvements. I offer the following comments for authors' consideration.

Please provide number of animals which were used for recording under LA2, LA5, and LL. The present study showed that the locomotor activity was diurnal. This is in contrast to previous studies that clearly showed nocturnal locomotor activity in the same species (e.g., Tomioka and Chiba, 1982, *J. Comp. Physiol.* 147:299-304; Ikeda and Tomioka, 1993, *Zool. Sci.*, 10: 597-604). It is better to discuss why the phase was reversed in this study. It may be better to describe how the authors determine the synchronization of the rhythms to

given LDs, because the daily rhythms are often caused by masking effects of light that are bypassing the endogenous clock.

L139: The authors stated that the rhythm was practically absent in LL treatment. This seems to be inadequate, because considerable number of animals showed rhythms under LL. Actually 17 and 24 animals were rhythmic under LL in stridulatory and locomotor activity, respectively.

According to Table S1, these are about 70% and 60%. Then why do the profiles in Figure 3A, B show arrhythmic pattern?

L144-145: a significant period should be better.

L145, 154: Why did you compare “medians” instead of means?

L146, 148, 154, 155 and 156: Please provide SD for each condition. This is important to know variance.

Figure 3C, D: What do colored areas mean? Please explain what black vertical lines are.

L175: Table S1 & Table S2. Table S2 is missing.

L194: reference #67 is missing in the list.

L213: “locomotion activity revealed some resilience to ALAN”: Is there any possibility that the observed synchronized locomotor rhythm is a masking effect of light?

L214: Please insert comma after exposure.

Decision letter (RSPB-2021-0233.R0)

09-Apr-2021

Dear Dr Ayali:

I am writing to inform you that your manuscript RSPB-2021-0233 entitled "Lifelong exposure to artificial light at night impacts stridulation and locomotion activity patterns in the cricket *Gryllus bimaculatus*" has, in its current form, been rejected for publication in Proceedings B.

This action has been taken on the advice of referees, who have recommended that substantial revisions are necessary. With this in mind we would be happy to consider a resubmission, provided the comments of the referees are fully addressed. However please note that this is not a provisional acceptance. Indeed, there was interest in the study but also uniformly strong critiques of / requests for clarification of the science. This sets a high bar for resubmission: all major concerns must be satisfied and our policy is not to allow multiple rounds of review. Please consider this carefully. Yet the science was considered exciting enough to merit further consideration.

4) Data - please see our policies on data sharing to ensure that you are complying (<https://royalsociety.org/journals/authors/author-guidelines/#data>).

Note that in review there were also concerns raised about data openness: "The raw data is missing, this is only the output of the models it seems. I also would like to see the measurements of the light sources. The spectrograms do give insight but can not be used to calculate other attributes like photon flux. With the data from the measurements that is possible."

Sincerely,
Dr John Hutchinson
mailto: proceedingsb@royalsociety.org

Associate Editor
Comments to Author:
Associate Editor: Doug Altshuler

This is an interesting study of the effects of light pollution at night on the movement of crickets. The referees are in agreement that that the authors have performed difficult experiments that are potentially very informative. However, this was a case in which the comments to the editor indicated a higher level of concern than will be apparent from the reviews. Thus, it is not clear to me their full set of concerns can be addressed with the current data set. I will summarize the more substantive private comments. If the authors wish to resubmit then, in addition, the stated referee concerns, they should also address the following:

- 1) There was considerable confusion about sample size. The stated numbers differ at different parts of the manuscript.
- 2) It was unclear how many light levels were used and whether these were applied to individuals or groups. If all individuals were reared together under one light condition then essentially $n=1$.
- 3) There is a mismatch between the statistical analyses and the stated hypotheses, both in terms of response to light and response via circadian rhythmicity.
- 4) The manuscript emphasizes lifelong exposure, but it was unclear if larval stages were exposed. The behavioral monitoring only took place during five days.

Reviewer(s)' Comments to Author:

Referee: 1

Comments to the Author(s)

Reviewer comments

Title: Lifelong exposure to artificial light at night impacts stridulation and locomotion activity patterns in the cricket

This study monitored behavior of crickets under different intensities of ALAN. I believe it should make a good contribution to the field of circadian and ALAN research. However, I have concerns about the lack of clarity of the methods and analyses that are outlined in more detail below.

Briefly, sample numbers, number of chambers (both rearing and experimental), etc are unclear in methods. Although light was applied at the adult stage, it is not a lifelong application if not for the larval stage and behavior monitoring only occurred during 5 days, therefore, I would be cautious about the use of "lifelong" in all places of the manuscript. Additionally, the analyses is comparing the same individual in paired T-tests, whereas different individuals were exposed to different treatments. In order to tell if light had an effect, treatment needs to be compared (see below). The circadian periodicity analyses also should be updated with circadian analyses rather than just mean and variance, that way you can specify shift, amplitude, etc.

Abstract: unclear why abstract was divided into different paragraphs

L25: explain these differences more clearly; same as in L27: the descriptions of differences in abstract is a bit vague

L37: is there a global statistic? Or is the cricket endemic to Europe?

L47: could you specify what exactly is missing?

Methods:

L77: were daylight and nightlight the same? All fluorescent?

L80: what does LA stand for? Why not use ALAN? Does the A mean less light? Could abbreviate to LLdim2lux?

In general through ms and also in title: it states that crickets were under lifelong exposure but it doesn't say at which point in their life cycle light treatment was applied. Also during the larval stage? At some point it says males 3-7 days post adult emergence, but that would mean they would still have at least 2 weeks to live but monitoring was only done for 5 days?

How many chambers were there? Did the crickets rotate in the chambers?

L89: says that 5W bulb was used in chambers, and that it was the same light as rearing light. But unclear earlier that type of rearing light it was.

How many rearing chambers were there? Were individual crickets reared in these chambers or grouped? It might be problematic if they were in groups and exposed to light as treatment is then by chamber and not by individual.

Either way, there needs to be a control (random effect) of rearing chamber (if more than one).

How many male crickets were used in total? How many were actually analyzed? First time sample numbers were evident was in L133. N=11 vs 15?

Why were crickets analyzed using paired T-tests with themselves? Shouldn't nocturnal activity be analyzed by treatment to see if treatment had an effect and used with rearing conditions as a random effect? Here you can only see if diurnal and nocturnal activity differed, and of course they did, in BOTH groups. To see if control and treatment conditions differed in nocturnal activity, you need a linear model with treatment as a fixed effect.

In figure 1, also unclear how data were normalized.

Based on FigS1, it's not just that the intensities that changed but also wavelengths, 2lux light seems to have more red wavelengths than other treatments. This should be discussed in the manuscript.

FS2 L141: were before weights and lengths measured? Can you do a body condition index based on single weight and length metrics rather than all of them?

L147: I don't understand how there is such a difference in sample numbers across different analyses, some had n=19, but then n=15, n=11. Could authors please have a table of sample numbers somewhere and explain the discrepancies?

L119: period analyses is more than just mean and variance but also cycles. Highly recommend using circadian analyses to look at circadian behavior. For example, Cosinor v.1.1 package in R (Barnett and Dobson, 2010) can be used to test treatment effects on amplitude (i.e. difference between peak and mean value of wave) and acrophase (i.e. time of peak expression in wave).

You can also use Rhythmicity Analysis Incorporating Nonparametric (RAIN) and Detection of Differential Rhythmicity (DODR) analyses to test for differences between the circadian rhythmicity and peak phase of control and ALAN groups (Thaben and Westermarck, 2016, 2014).

Fig4: it would be nice to know which ALAN treatment resulted in which behaviors.

L200: I feel that some of the explanations of the importance of these behaviors should be in the intro rather than the discussion

Referee: 2

Comments to the Author(s)

This is a very nice paper that shows an impact of ALAN that was previously unknown. ALAN clearly affects the timing of the behavior of crickets and the impact increases with intensity.

Especially that the two behaviors studied seem to respond independently is important as this indicates that ALAN can result in desynchronization of different behaviors. The results are very convincing in that perspective.

One aspect that deserves more attention is the choice of light levels. It is shortly discussed that the diurnal light level is “relatively low” with 40 lux. A typical summer day is 2000 (very cloudy) to >100.000 lux. A value of 40 lux is overcast at dusk. I understand the practical objections to high light levels however and this is made explicit. The 2 and 5 lux levels are reasonable. These are levels you find near illumination and in urban areas. It would be interesting to know the impact of lower (sky-glow) levels but that is a different question. However, the control treatment is 0 lux. This raises a few questions. The first is how do you monitor movement without any light source? Or was there an IR-source? More importantly 0 lux is far from a natural control. These crickets are active at night and use the low levels of natural nocturnal light to see. Having true darkness deprives them of sight. So the control that should reflect the natural condition, high light level (>1000 lux) during the day and low light levels (0.0001-0.05 lux) at night, is low light levels during the day and unnatural darkness at night. This might very well affect the activity of the crickets.

The presentation of the light treatment in the materials and methods is also very limited. If only one type of light source was used how were the different light levels achieved? How did you measure these, where (middle, or multiple spots) and with which device? It is written that the CFL bulb emits 40 lux, the appropriate unit for luminous flux is lumen, not lux. Or did you measure illumination at the bottom of the cage? In that case it is only a part of what the bulb emits but in lumen (and more relevant).

The blue part of the spectra differs quite substantially between the light levels, this might be relevant for entrainment and warrants discussion. I would also suggest giving the spectral measurements in Dryad, this allows others to calculate different aspects of your treatment (photon flux, intensity based on the spectral sensitivity of the eyes of crickets etc.), a graph gives insight but does not allow for further analysis. It would also be helpful to upload the raw data with labels that are easily interpretable.

Referee: 3

Comments to the Author(s)

This is an interesting paper dealing with the effect of artificial light exposure at night (ALAN) on daily rhythms of stridulatory and locomotor activities. The authors showed that ALAN had significant and differential effects on the two rhythms, suggesting that ALAN affects the behavioral rhythms in this species and that the two rhythms were controlled by separate mechanisms. The experiments were appropriately performed, the results are interesting and discussed reasonably. However, the manuscript needs some improvements. I offer the following comments for authors' consideration.

Please provide number of animals which were used for recording under LA2, LA5, and LL. The present study showed that the locomotor activity was diurnal. This is in contrast to previous studies that clearly showed nocturnal locomotor activity in the same species (e.g., Tomioka and Chiba, 1982, *J. Comp. Physiol.* 147:299-304; Ikeda and Tomioka, 1993, *Zool. Sci.*, 10: 597-604). It is better to discuss why the phase was reversed in this study.

It may be better to describe how the authors determine the synchronization of the rhythms to given LDs, because the daily rhythms are often caused by masking effects of light that are bypassing the endogenous clock.

L139: The authors stated that the rhythm was practically absent in LL treatment. This seems to be inadequate, because considerable number of animals showed rhythms under LL. Actually 17 and 24 animals were rhythmic under LL in stridulatory and locomotor activity, respectively.

According to Table S1, these are about 70% and 60%. Then why do the profiles in Figure 3A, B show arrhythmic pattern?

L144-145: a significant period should be better.

L145, 154: Why did you compare “medians” instead of means?

L146, 148, 154, 155 and 156: Please provide SD for each condition. This is important to know variance.

Figure 3C, D: What do colored areas mean? Please explain what black vertical lines are.

L175: Table S1 & Table S2. Table S2 is missing.

L194: reference #67 is missing in the list.

L213: "locomotion activity revealed some resilience to ALAN": Is there any possibility that the observed synchronized locomotor rhythm is a masking effect of light?

L214: Please insert comma after exposure.

Author's Response to Decision Letter for (RSPB-2021-0233.R0)

See Appendix A.

RSPB-2021-1626.R0

Review form: Reviewer 3

Recommendation

Accept with minor revision (please list in comments)

Scientific importance: Is the manuscript an original and important contribution to its field?

Excellent

General interest: Is the paper of sufficient general interest?

Excellent

Quality of the paper: Is the overall quality of the paper suitable?

Good

Is the length of the paper justified?

Yes

Should the paper be seen by a specialist statistical reviewer?

No

Do you have any concerns about statistical analyses in this paper? If so, please specify them explicitly in your report.

No

It is a condition of publication that authors make their supporting data, code and materials available - either as supplementary material or hosted in an external repository. Please rate, if applicable, the supporting data on the following criteria.

Is it accessible?

N/A

Is it clear?

N/A

Is it adequate?

N/A

Do you have any ethical concerns with this paper?

No

Comments to the Author

I appreciate the authors' response to my previous concerns. They have responded to all of them appropriately. However, I found one point that is difficult to understand in the newly added part. The authors calculated diurnal and nocturnal activity for both stridulation and locomotion under various lighting regimens. But it seems very difficult to determine day and night phases for arrhythmic animals in LL. The authors should describe how they calculated diurnal and nocturnal activity in LL animals in Materials and Methods section.

L172: I think LL5 should read LL2.

Review form: Reviewer 4

Recommendation

Accept with minor revision (please list in comments)

Scientific importance: Is the manuscript an original and important contribution to its field?

Excellent

General interest: Is the paper of sufficient general interest?

Good

Quality of the paper: Is the overall quality of the paper suitable?

Excellent

Is the length of the paper justified?

Yes

Should the paper be seen by a specialist statistical reviewer?

No

Do you have any concerns about statistical analyses in this paper? If so, please specify them explicitly in your report.

No

It is a condition of publication that authors make their supporting data, code and materials available - either as supplementary material or hosted in an external repository. Please rate, if applicable, the supporting data on the following criteria.

Is it accessible?

Yes

Is it clear?

Yes

Is it adequate?

Yes

Do you have any ethical concerns with this paper?

No

Comments to the Author

Please see attached file. (See Appendix B)

Decision letter (RSPB-2021-1626.R0)

23-Aug-2021

Dear Dr Ayali

I am pleased to inform you that your manuscript RSPB-2021-1626 entitled "Lifelong exposure to artificial light at night impacts stridulation and locomotion activity patterns in the cricket *Gryllus bimaculatus*" has been accepted for publication in Proceedings B. Congratulations!!

The referee(s) have recommended publication, but also suggest some minor revisions to your manuscript. Therefore, I invite you to respond to the referee(s)' comments and revise your manuscript. Because the schedule for publication is very tight, it is a condition of publication that you submit the revised version of your manuscript within 7 days. If you do not think you will be able to meet this date please let us know.

There are some constructive critiques that will take some moderate effort to resolve.

Sincerely,

Dr John Hutchinson

Editor

Associate Editor

Comments to Author:

Associate Editor: Doug Altshuler

Ayali et al. have revised their manuscript on light pollution and cricket locomotion according to three very constructive reviews. Unfortunately, two of the original referees were not available to evaluate the revision, but we have obtained a fourth referee that also (like me) considered responses to the other reviews. We are in consensus that this revision has been responsive to the earlier comments, and that the manuscript presents an interesting and well executed study. There are a handful of minor comments that should be addressed.

Reviewer(s)' Comments to Author:

Referee: 3

Comments to the Author(s).

I appreciate the authors' response to my previous concerns. They have responded to all of them appropriately. However, I found one point that is difficult to understand in the newly added part. The authors calculated diurnal and nocturnal activity for both stridulation and locomotion under various lighting regimens. But it seems very difficult to determine day and night phases for arrhythmic animals in LL. The authors should describe how they calculated diurnal and nocturnal activity in LL animals in Materials and Methods section.

L172: I think LL5 should read LL2.

Referee: 4

Comments to the Author(s).

Please see attached file.

Author's Response to Decision Letter for (RSPB-2021-1626.R0)

See Appendix C.

Decision letter (RSPB-2021-1626.R1)

31-Aug-2021

Dear Dr Ayali

I am pleased to inform you that your manuscript entitled "Lifelong exposure to artificial light at night impacts stridulation and locomotion activity patterns in the cricket *Gryllus bimaculatus*" has been accepted for publication in Proceedings B.

Data Accessibility section

Open Access

Paper charges

Sincerely,

Proceedings B

Appendix A

Dear Editor,

We greatly appreciate the in-principle positive assessment of our manuscript (ID RSPB-2021-0233) by the editors and reviewers and are grateful for all the constructive criticisms and suggestions for improvement. We have followed very thoroughly all the comments, and accordingly, as presented in much details below, made substantial changes to our paper in order to resolve all uncertainties and unclarities.

Specifically, we are grateful to the reviewers for pointing out the weaknesses in the parts of our work related to the description and details of the methodology used, and we have therefore put much emphasis on improving this section. The revision also includes further analysis performed following the requests of the referees, resulting in additional new figures, a new supplementary figure and table.

We believe all these changes have much improved the paper, and hope you will find it ready for publication in Proceedings B.

Best Regards,

Amir Ayali

Reviewer(s)' Comments to Author:

Referee: 1

Comments to the Author(s)

Reviewer comments

1. This study monitored behavior of crickets under different intensities of ALAN. I believe it should make a good contribution to the field of circadian and ALAN research. However, I have concerns about the lack of clarity of the methods and analyses that are outlined in more detail below. Briefly, sample numbers, number of chambers (both rearing and experimental), etc are unclear in methods. Although light was applied at the adult stage, it is not a lifelong application if not for the larval stage and behavior monitoring only occurred during 5 days, therefore, I would be cautious about the use of “lifelong” in all places of the manuscript. Additionally, the analyses is comparing the same individual in paired T-tests, whereas different individuals were exposed to different treatments. In order to tell if light had an effect, treatment needs to be compared (see below). The circadian periodicity analyses also should be updated with circadian analyses rather than just mean and variance, that way you can specify shift, amplitude, etc.

Response and revisions made:

We wish to thank the referee for the thorough reading of our manuscript and for this summary and overall positive assessment of our work. The constructive feedback, specifically regarding insufficient detailed description of the methods and samples sizes is greatly appreciated.

All the experimental insects have been subjected to the specific tested light conditions from hatching, through all larval stadia, to the adult stage, including the days of the actual experiment. Behavioral

monitoring was conducted on male crickets transferred individually from their rearing cage to the anechoic chamber (with no change in light conditions). Hence, we refer to the different light regimes as *lifelong*.

We thank the referee for recommending to include in the MS further analyses, such as treatment comparison and the body condition index, which were conducted but not included previously. We now also include the circadian phase analyses, which we hope have further improved the presentation of our findings.

2. Abstract: unclear why abstract was divided into different paragraphs

Response and revisions made: Corrected. The abstract is now one concise paragraph.

3. L25: explain these differences more clearly; same as in L27: the descriptions of differences in abstract is a bit vague

Response and revisions made: We appreciate this comment and have rephrased the text to be clearer. Please see L24 – L30: “In contrast, ALAN affected both the relative level and timing of the crickets’ nocturnal and diurnal activity. ALAN induced free-running patterns, manifested in significant changes in the median and variance of the activity periods, and even arrhythmic behavior. The effects were mostly light-intensity-dependent, revealing an increase in the difference between the activity periods calculated for stridulation and locomotion in the same individual. This finding may indicate the existence of two peripheral clocks.”

4. L37: is there a global statistic? Or is the cricket endemic to Europe?

Response and revisions made: The cricket *Gryllus bimaculatus* is indeed not endemic to Europe, therefore we have replaced the European statistics with the relevant global data. L37 now read: “with more than 80% of the world population living under light-polluted skies”

5. L47: could you specify what exactly is missing?

Response: We thank the referee for raising this point and now add the missing information.

Revisions: L46 – L48: “However, our knowledge of the effects of ALAN, specifically of its various intensities, on insect behavior, temporal activity partitioning, and fitness, is far from complete.”

Methods:

6. L77: were daylight and nightlight the same? All fluorescent?

Response: We thank the reviewer for pointing this out. Indeed, all light bulbs used in both, growth chambers and experimental chambers, were compact fluorescent light bulbs from the same company (Nepton) and the same spectral parameters (6500 K, 380-780 nm, peak: 547 & 612 nm). Daylight and nightlight bulbs in the growth chamber and the experimental setup differed in energy (Watt) only. ALAN intensities were achieved by shading the light bulbs.

Revisions:

L82 – L87: “*G. bimaculatus* crickets were reared under a constant temperature of $26\pm 2^{\circ}\text{C}$. Different experimental groups were reared from the egg stadium to hatching and through all life stages (i.e. lifelong) under one of four different light regimes. The rearing chambers were illuminated with white fluorescent light (CFL, NeptOn, 6500 K, 380-780 nm, peak: 547 & 612 nm; different light intensities were achieved by shading the light bulbs).”

L99 – L107: “Light was provided via a 5W white CFL bulb (NeptOn, 6500 K, 380-780 nm, peak: 547 & 612 nm, see Fig. S1). Lighting conditions and the light regimes in the experimental chamber were similar to those that the insect had experienced in the rearing chambers (one of four, as described above). Particular care was taken to randomize the group-experimental chamber linkage. Light intensities were measured at four locations at the bottom of the chamber. Measurements were conducted at the cricket’s eye level, at a distance of approx. 65cm from the light bulb, using a digital light meter (TES-1337, TES, Taiwan). The light spectra were recorded using a Sekonic Spectromaster C-700 (North White Plains, NY, USA).”

7. L80: what does LA stand for? Why not use ALAN? Does the A mean less light? Could abbreviate to LLdim2lux?

Response: Indeed, it is more common to indicate nocturnal illumination using LLdim, together with the appropriate intensities. However, we felt a shorter acronym may be better and more useful here. LA stood for light (L) and for ALAN (A). Following the referee’s comment, we have changed this. Now LD stands for light-dark and LL for constant 40 lux light, while nocturnal dim light is indicated with the appropriate intensities (LL2 lux, LL5 lux). We hope that the new abbreviations are clear.

Revisions:

All LA2 and LA5 were replaced by LL2 and LL5 throughout the manuscript.

L87 – L90: “Conditions between groups varied during the 12h night period, as follows: (1) 12h daylight:12h dark (LD, control), (2) 12h daylight:12h 2lux ALAN (LL2), (3) 12h daylight:12h 5lux ALAN (LL5), and (4) 24h constant daylight (LL).”

8. In general through ms and also in title: it states that crickets were under lifelong exposure but it doesn’t say at which point in their life cycle light treatment was applied. Also during the larval stage? At some point it says males 3-7 days post adult emergence, but that would mean they would still have at least 2 weeks to live but monitoring was only done for 5 days?

Response: This issue was indeed not clear enough in the manuscript and we thank the referee for noticing this. The crickets were submitted to lifelong light conditions, from hatching, through all larval stadia, to the adult stage, including the days of the actual experiment. As the referee correctly noted, the experiments themselves were conducted on adult males 3-7days post adult emergence. Our novel setup allowed continuous, simultaneous monitoring of locomotion and stridulation behavior. However, for technical reasons, and given the complexity of the setup, the duration of the experiments was limited. Different experiments were actually 5-12 days long, and 5 days were chosen as the longest common period for the analyses of means. We agree with the referee that this time period may be considered somewhat on the short side, however, we found it to be sufficient for representing the individual behaviors and the light effects.

Revisions:

L82 - L84: “Different experimental groups were reared from the egg stadium to hatching and through all life stages (i.e. lifelong) under one of four different light regimes.”

9. How many chambers were there? Did the crickets rotate in the chambers?

Response: We thank the referee for pointing out this unclarity. We custom built four identical anechoic chambers, aka. experimental systems. Individual crickets from the different experimental groups were assigned to a chamber randomly.

Revisions:

L94 – L97: “Different experimental groups were reared from the egg stadium to hatching and through all life stages (i.e. lifelong) under one of four different light regimes. (Fig. 1).”

L102 – L103: “Particular care was taken to randomize the group-experimental chamber linkage.”

10. L89: says that 5W bulb was used in chambers, and that it was the same light as rearing light. But unclear earlier that type of rearing light it was.

Response: We thank the referee for pointing out this missing information and now add the description of the light bulb and shading methods.

Revisions:

L84 – L87: “The rearing chambers were illuminated with white fluorescent light (CFL, NeptOn, 6500 K, 380-780 nm, peak: 547 & 612 nm; different light intensities were achieved by shading the light bulbs).”

L99 – L102: “Light was provided via a 5W white CFL bulb (NeptOn, 6500 K, 380-780 nm, peak: 547 & 612 nm, see Fig. S1). Lighting conditions and the light regimes in the experimental chamber were similar to those that the insect had experienced in the rearing chambers (one of four, as described above).”

11. How many rearing chambers were there? Were individual crickets reared in these chambers or grouped? It might be problematic if they were in groups and exposed to light as treatment is then by chamber and not by individual. Either way, there needs to be a control (random effect) of rearing chamber (if more than one).

Response: The crickets were raised in breeding colonies housed in adjacent similar rearing chambers. All rearing conditions but the light regimes were kept identical. The experimental animals were raised under the same light treatment later applied to the experimental anechoic chambers. There were four anechoic chambers, aka. experimental systems, located in the same room and submitted to the same conditions, but the appropriate lighting regime. Individual male crickets, aged 3-7 days post adult emergence, were removed from the rearing chamber and randomly assigned an anechoic chamber. Hence, n=number of individuals monitored.

Revisions:

Please see our revision to comment 9.

L94 – L97, and L102 – L103.

12. How many male crickets were used in total? How many were actually analyzed? First time sample numbers were evident was in L133. N=11 vs 15?

Response: This is an important point and we wish to thank the referee for the opportunity to clarify the sample sizes used for different calculations, and mostly the limitations and rationale leading to different sample sizes throughout our work. The methods describe a setup enabling simultaneous, continuous monitoring. Still, due to technical reasons sometimes experiments ended with stridulation or locomotion data only. These experiments were used to calculate the results presented in figure 2 and 3, as well as the percentages of the different rhythms (figure 5B1&2). For the analysis presented in figure 4, only significant period values could be used, whereas all arrhythmic individuals had to be removed from the analysis, leading to a new, smaller sample size, representing the population. In a third step, we conducted individual rhythms comparisons and correlation analysis between stridulation and locomotion behavior, which meant we could analyze only animals for which both periods were found numeric and significant, resulting in further removal of some individuals and a third, somewhat different, sample size. Following this comment, together with comments made by the other referees,

and in order to avoid confusion, we have now added a new supplementary table S1, presenting and clarifying the different sample sizes and categories.

Revisions:

L168, L174, L178, L181, L189, L216, L224, L244: "Table S1"

Supplementary Table S1: Sample sizes of individual male *Gryllus bimaculatus* adult crickets used for each analysis and treatment of the lifelong ALAN experiment. Each column represents another set of analysis on stridulation (left part) or locomotion (right part) behavior. Patterns & percentages include all experimental animals. For the period and acrophase analysis, sample size includes only animals, which period analysis was found significant, hence, all individuals with a synchronized or free-run pattern (but no arrhythmic results). The sample size for the correlation included only individuals, for each both, stridulation and locomotion period results were obtained from the same animal."

13. Why were crickets analyzed using paired T-tests with themselves? Shouldn't nocturnal activity be analyzed by treatment to see if treatment had an effect and used with rearing conditions as a random effect? Here you can only see if diurnal and nocturnal activity differed, and of course they did, in BOTH groups. To see if control and treatment conditions differed in nocturnal activity, you need a linear model with treatment as a fixed effect.

Response: Crickets from the control group (LD) were analyzed using paired t-tests to compare the diurnal and nocturnal behavior of the same individuals (see L164 - L168). Following the reviewer comments, we extended our analysis and conducted a quantitative analysis comparing separately diurnal, and nocturnal activity in the different experimental groups. We found that light treatment had an effect on nocturnal and diurnal activity. The effect, however, was caused by different sources in both behavioral activities. In stridulation, the LD treatment significantly differed from LL5 and LL treatments while in locomotion, the LL treatment significantly differed from all three other treatments. This new result was found to be in accord with our conclusions concerning the different susceptibility and resilience (or masking effect (L313 – L314)) of stridulation and locomotion to ALAN. Following the referee's comment, we have added figure 3 to the manuscript. We believe that the description of the activity under ALAN is now significantly improved and wish to thank the referee for prompting us to deepen our analysis.

Revisions:

Figure 3: "ALAN intensities induce variation in the temporal activity and rhythmicity of crickets. Quantitative comparisons of diurnal (yellow, small letters) and nocturnal (black and grey, capital letters) stridulation (A1) and locomotion (B1) activity under LD (n=15, n=11, respectively), LL2 (n=20, n=25, respectively), LL5 (n=20, n=17, respectively) and LL (n=21, n=40, respectively) treatments (Mean±SE). Different letters represent statistical differences. The differences between nocturnal and diurnal activity of stridulation (A2) and locomotion (B2) under LD (blue), LL2 (purple), LL5 (green) and LL (red) treatments (Mean±SE)."

L137 – L138: "Comparisons of nocturnal and diurnal activity, as well as the absolute difference within groups, were conducted with the Kruskal Wallis test, followed using a Dunn's test."

L170 – L181: "A quantitative comparison of diurnal and nocturnal activity in the different experimental groups revealed that the median of the normalized diurnal, as well as nocturnal LD stridulation activity level significantly differed from the LL5 and LL treatments ($p < 0.007$ for all, Kruskal-Wallis test with Dunn's multiple comparisons, Fig 3A1, Table S1). Significant differences in diurnal locomotor activity level were found between LL and all other treatments, while for the nocturnal locomotor activity levels, differences were significant only between LL and both LL2 and LL5 ($p < 0.01$ for diurnal locomotion, $p < 0.02$ for nocturnal locomotion, Kruskal-Wallis test with Dunn's multiple comparisons, Fig 3B1, Table S1).

In both behaviors, the effects of the light regime and ALAN were also apparent when comparing the difference between diurnal and nocturnal activity levels. (Kruskal-Wallis test, $p < 0.0001$, Fig 3A2, 3B2, Table S1).”

14. In figure 1, also unclear how data were normalized.

Response: We have revised the y-axis of figure 2 in accord with figure 3, added the normalization to the legends and rephrased the normalization in the MS.

Revisions:

L130 - L135: “For diurnal and nocturnal quantitative activity comparisons, values were normalized for each individual by dividing that individual’s values by its own mean value. For behavior, rhythmicity, and period analyses and comparisons, values were normalized for each individual by dividing that individual’s values by its own maximum value, resulting in an activity index ranging from 0 (no activity) to 1 (maximal activity).”

Figure 2: The y-axis now reads “Stridulation events/h (Mean \pm SE)” and “Locomotion events/h (Mean \pm SE)”.

Figure legend 2: “Values were normalized by dividing each individual’s values by its own maximum value, resulting in an activity index ranging from 0 (no activity) to 1 (maximal activity).”

Figure legend 3: “Values for A1,2 & B1,2 were normalized by dividing each individual’s values by its own mean value, while values for A3 & B3 were normalized by dividing each individual’s values by its own maximum value, resulting in an activity index ranging from 0 (no activity) to 1 (maximal activity).”

15. Based on FigS1, it’s not just that the intensities that changed but also wavelengths, 2lux light seems to have more red wavelengths than other treatments. This should be discussed in the manuscript.

Response: We thank the referee for noticing and bringing this change in wavelength to our attention. This comment and the comments of referee 2, have prompted us to revisit the relevant literature. We have added a section discussing the spectral sensitivity of cricket photoreceptors, based on Zufall, et al., 1989 and on the performance of these receptors (Frolov, et al., 2014). We do conclude, based on these papers, that the model cricket, *Gryllus bimaculatus*, has a spectral sensitivity of up to 600nm and does not, therefore, see the rise in intensities of the red wavelengths (>700nm).

Revisions:

L274 – L 284: “When studying the effects of light on behavior, the visual system of the model insect should be taken into consideration. The following spectral types have been described in the cricket *G. bimaculatus*: UV (peak: 332 nm), blue (peak: 445 nm), and green (peak: 515 nm), resulting in a spectral sensitivity of up to 600 nm [49], with the performance of the photoreceptors leading to the conclusion that the crickets’ eye has evolved for signal processing in dim light [50]. In our experiments, shading the light bulbs in order to create lower light intensities resulted in a relative decrease in the blue wavelength (400-500 nm) intensity and an increase in the red wavelength (>700 nm) intensity. Interestingly, while this increase in red wavelength had no influence on the experiments, as crickets do not possess the ability to see red light, a clear effect of LL2 was observed despite the decrease in the blue wavelength.”

16. FS2 L141: were before weights and lengths measured? Can you do a body condition index based on single weight and length metrics rather than all of them?

Response: Prompted by the referees' comment, we have now calculated the body condition index based on each single cricket's weight divided by its body length. Here, too, similar to the morphometric comparison in figure S2, an inconsistent picture emerged. The treatments per se were found to significantly affect the body condition index (Kruskal-Wallis, $p < 0.038$), yet Dunn's post-hoc test found no significant differences among any two of the different treatments (Dunn's multiple comparison, $p > 0.9$ for all comparisons).

Revisions:

Supplementary Figure S2: "Body condition index was added "Inserted: Body condition index of the ratio of the crickets' individual weights divided by their individual lengths and compared between all four experimental groups LD, LL2, LL5 and LL ($n=11, 22, 21,$ and $34,$ respectively), are presented."

L158– L162: "The lifelong lighting conditions had a somewhat inconsistent effect on the crickets' morphological characteristics (size, weight, and morphometric relations, Fig. S2). While we observed a significant effect on the body condition index (Kruskal-Wallis, $p < 0.038$), no significant differences were found among the different treatments (Dunn's post-hoc multiple comparison, $p > 0.9$ for all comparisons, Fig. S2)."

17. L147: I don't understand how there is such a difference in sample numbers across different analyses, some had $n=19$, but then $n=15, n=11$. Could authors please have a table of sample numbers somewhere and explain the discrepancies?

Response and revisions: We are grateful to the referee for encouraging us to clarify the issue of sample size numbers throughout our work. We have now added the supplementary table S1. Please see our revision in reply to comment 12 (L168, L174, L178, L181, L189, L216, L224, L244, and supplementary Table S1).

18. L119: Period analysis is more than just mean and variance but also cycles. Highly recommend using circadian analyses to look at circadian behavior. For example, Cosinor v.1.1 package in R (Barnett and Dobson, 2010) can be used to test treatment effects on amplitude (i.e. difference between peak and mean value of wave) and acrophase (i.e. time of peak expression in wave). You can also use Rhythmicity Analysis Incorporating Nonparametric (RAIN) and Detection of Differential Rhythmicity (DODR) analyses to test for differences between the circadian rhythmicity and peak phase of control and ALAN groups (Thaben and Westermark, 2016, 2014).

Response: We thank the referee for prompting us to conduct phase analysis and for recommending the packages for circadian analysis. We have followed the referee's suggestion and have calculated the acrophase for stridulation and locomotion behavior, for all treatments, using the CosinorPy package (Moškón, M., 2020) for Python. The acrophase calculations were followed by statistical analysis using Oriana program (Kovach, V.W., 2011). The analysis presented a certain resilience of the acrophase in relation to the various activity periods. However, we observed an overall trend in which the data distribution and variance grew with the light intensity. A comparison of stridulation and locomotion phases in the same individuals revealed a significant difference between the phases in the control, that was lost in the LL₅ and LL. We have now added a graph showing the circular data to fig. 4C&D and the related results were added to the appropriate section, as well as the numeric results in a supplementary table S3. We believe that the acrophase analysis, specifically the comparison in the same individual, has strengthened our conclusions and overall improved and the manuscript.

Revisions:

Methods: L149 – L155

Results: L223 – L249

Discussion: L305 – 309; L324 – L327

Figure 4: "ALAN affects the acrophase of individuals' behavior, and the group variance in stridulation (C) and locomotion (D) behavior. Each point in the circular plots represents the mean phase of five days of activity of an individual male cricket. Grid lines = 0.25. The black arrow represents the mean vector of phase, calculated for the entire experimental group."

Supplementary table S3: "Acrophase analyses of stridulation and locomotion activity of individual crickets from the four experimental treatments. Only animals whose activity period was found significant were included in the acrophase analysis. Each datapoint represents the acrophase calculated for five days. The difference data was calculated by subtracting locomotion values from stridulation values of the same individual. Data are presented for all individuals, and separately for those cases where data for both behaviors was available for the same individual."

19. Fig4: it would be nice to know which ALAN treatment resulted in which behaviors. (Now figure 5)

Response: Unfortunately, we failed to understand the meaning of the referee in this comment. In figure 5, panel A shows the types of behaviors, and panel B shows the resulted percentage of each behavior induced by each treatment.

20. L200: I feel that some of the explanations of the importance of these behaviors should be in the intro rather than the discussion

Response and revisions: We agree. We have now added this explanation to the introduction, see lines L67 – L70: "Locomotion is important for the individual's fitness, especially in the context of foraging and the risk of predation. Stridulation, explicitly calling songs used for communication and female attraction, is of course crucial for the reproduction of the species. Any asynchrony in these behaviors among the population is critical."

Referee: 2

Comments to the Author(s)

1. This is a very nice paper that shows an impact of ALAN that was previously unknown. ALAN clearly affects the timing of the behavior of crickets and the impact increases with intensity. Especially that the two behaviors studied seem to respond independently is important as this indicates that ALAN can result in desynchronization of different behaviors. The results are very convincing in that perspective.

Response:

We wish to thank the referee for this concise summary of our contribution, for the thorough reading of our manuscript, and for the insightful comments. We were excited to see the overall appreciative tone of the results presented. We welcome the constructive comments and discussion on the use of different light intensities and spectra and into the visual sensitivity of our model insect and agree on its importance. We hope the referee will find the revised manuscript clearer and improved.

2. One aspect that deserves more attention is the choice of light levels. It is shortly discussed that the

diurnal light level is “relatively low” with 40 lux. A typical summer day is 2000 (very cloudy) to >100.000 lux. A value of 40 lux is overcast at dusk. I understand the practical objections to high light levels however and this is made explicit. The 2 and 5 lux levels are reasonable. These are levels you find near illumination and in urban areas. It would be interesting to know the impact of lower (sky-glow) levels but that is a different question. However, the control treatment is 0 lux. This raises a few questions. The first is how do you monitor movement without any light source? Or was there an IR-source? More importantly 0 lux is far from a natural control. These crickets are active at night and use the low levels of natural nocturnal light to see. Having true darkness deprives them of sight. So the control that should reflect the natural condition, high light level (>1000 lux) during the day and low light levels (0.0001-0.05 lux) at night, is low light levels during the day and unnatural darkness at night. This might very well affect the activity of the crickets.

Response: We thank the referee for pointing out these substantial challenges in ALAN research, specifically regarding the choice of experimental day- and night-light intensities. We agree with the referee that the chosen conditions and light intensities do not faithfully represent a typical summer day nor the natural light conditions. Furthermore, in the laboratory the lights went on or off at once, lacking the natural gradual processes of sunrise and sunset and the wavelength-dependent changes in intensity. This is, indeed, an important topic to which we have dedicated much thought. The field cricket is known to live in borrows, and calls for mate at sunset. Accepted knowledge of the cricket’s natural history has led us to choose relatively low daylight intensities, also in order to avoid visual discomfort to the insects as well as to avoid overheating of the anechoic chamber. We agree with the referee that extremely low nocturnal light levels may better reflect natural night conditions than complete darkness. Nevertheless, we sought to contrast the low ecologically relevant ALAN treatments of 2 and 5 lux with both the diurnal and nocturnal conditions. We can’t exclude that some of the observed behavioral patterns (e.g. free-run) may have been triggered by these differences in intensity in the diurnal and nocturnal light, and we have, therefore, already initiated a follow-up outdoor experiment in order to provide more insights into ALAN effects on cricket behavior under more natural conditions. We also agree with the referee that it would be interesting to look at the impact of sky-glow levels on crickets’ behavior, especially as they do see polarized light and sky-glow may affect vision-based orientation. We have now added to the MS an explanation regarding the use of an Infra-red camera in order to enable detections during both, light- and scotophases and rephrased the relevant sentences in the discussion.

Revisions:

L118 – L119: “IR enabled tracking the animals’ locomotion under all light regimes, including scotophases.”

L328 – L333: “Our experiments were conducted under laboratory conditions, with a relatively low daylight intensity in order to avoid both high chamber temperatures and visual discomfort to the insects (which live a cryptic life). The chosen light intensity may not faithfully represent natural light conditions. Furthermore, the experimental illuminations were turned on and off abruptly, lacking the natural gradual processes of sunrise and sunset.”

3. The presentation of the light treatment in the materials and methods is also very limited. If only one type of light source was used how were the different light levels achieved? How did you measure these, where (middle, or multiple spots) and with which device? It is written that the CFL bulb emits 40 lux, the appropriate unit for luminous flux is lumen, not lux. Or did you measure illumination at the bottom of the cage? In that case it is only a part of what the bulb emits but in lumen (and more relevant).

Response: Indeed, only one type of light source was used and the different intensities were achieved by shading, i.e. covering the light bulb with porous felt. In all chambers, light measurements took place at

four different spots, at the bottom of the anechoic chamber, on cricket eye-level. We thank the referee for his/her correction, about not measuring the bulb's emission in lumen, but the light intensity on the surface, in lux, at approximately 65cm distance from the light source. It is now corrected in the MS.

Revisions:

L103 – L107: "Light intensities were measured at four locations at the bottom of the chamber. Measurements were conducted at the cricket's eye level, at a distance of approx. 65cm from the light bulb, using a digital light meter (TES-1337, TES, Taiwan). The light spectra were recorded using a Sekonic Spectromaster C-700 (North White Plains, NY, USA)."

4. The blue part of the spectra differs quite substantially between the light levels, this might be relevant for entrainment and warrants discussion. I would also suggest giving the spectral measurements in Dryad, this allows others to calculate different aspects of your treatment (photon flux, intensity based on the spectral sensitivity of the eyes of crickets etc.), a graph gives insight but does not allow for further analysis. It would also be helpful to upload the raw data with labels that are easily interpretable.

Response: We thank the referee for the suggestion to provide the spectral data in Dryad and have added both measurements of the IR-camera, and of the 5W CFL light bulb to Dryad. Unfortunately, the portable measuring devices capable of measuring the light intensity in the anechoic chamber (and hence at the cricket eye-level) provide either lux values only (TES-1337, TES, Taiwan) or graphic, relative spectral emittance (Sekonic Spectromaster C-700, North White Plains, NY, USA), as shown in supplementary figure S1, but no photon flux numbers. Nevertheless, this comment and comment 15 of referee 1, have prompted us to revisit the relevant literature. We have added a section discussing the spectral sensitivity of cricket photoreceptors, based on Zufall, et al., 1989 and on the performance of these receptors (Frolov, et al., 2014).

Revisions:

L274 – L284: "When studying the effects of light on behavior, the visual system of the model insect should be taken into consideration. The following spectral types have been described in the cricket *G. bimaculatus*: UV (peak: 332 nm), blue (peak: 445 nm), and green (peak: 515 nm), resulting in a spectral sensitivity of up to 600 nm [49], with the performance of the photoreceptors leading to the conclusion that the crickets' eye has evolved for signal processing in dim light [50]. In our experiments, shading the light bulbs in order to create lower light intensities resulted in a relative decrease in the blue wavelength (400-500 nm) intensity and an increase in the red wavelength (>700 nm) intensity. Interestingly, while this increase in red wavelength had no influence on the experiments, as crickets do not possess the ability to see red light, a clear effect of LL2 was observed despite the decrease in the blue wavelength."

Referee: 3

Comments to the Author(s)

1. This is an interesting paper dealing with the effect of artificial light exposure at night (ALAN) on daily rhythms of stridulatory and locomotor activities. The authors showed that ALAN had significant and differential effects on the two rhythms, suggesting that ALAN affects the behavioral rhythms in this species and that the two rhythms were controlled by separate mechanisms. The experiments were appropriately performed, the results are interesting and discussed reasonably. However, the manuscript needs some improvements. I offer the following comments for authors' consideration.

Response: We wish to thank the referee for this clear and appreciative summary of our work, for the thoughtful reading of our manuscript, and the constructive comments. We specifically are grateful for pointing out the effects that can be explained with masking and for prompting us to refine our conclusions and figure legends. We hope the referee will find the revised manuscript clearer and improved.

2. Please provide number of animals which were used for recording under LL2, LL5, and LL.

Response: We thank the reviewer for encouraging us to clarify the sample size throughout the work and for this purpose added the supplementary table S1 to the manuscript.

Revisions:

L168, L174, L178, L181, L189, L216, L224, L244: "Table S1"

Supplementary Table S1: Sample sizes of individual male *Gryllus bimaculatus* adult crickets used for each analysis and treatment of the lifelong ALAN experiment. Each column represents another set of analysis on stridulation (left part) or locomotion (right part) behavior. Patterns & percentages include all experimental animals. For the period and acrophase analysis, sample size includes only animals, which period analysis was found significant, hence, all individuals with a synchronized or free-run pattern (but no arrhythmic results). The sample size for the correlation included only individuals, for each both, stridulation and locomotion period results were obtained from the same animal."

3. The present study showed that the locomotor activity was diurnal. This is in contrast to previous studies that clearly showed nocturnal locomotor activity in the same species (e.g., Tomioka and Chiba, 1982, *J. Comp. Physiol.* 147:299-304; Ikeda and Tomioka, 1993, *Zool. Sci.*, 10: 597-604). It is better to discuss why the phase was reversed in this study.

Response: This point, indeed, deserves more careful attention and we thank the referee for mentioning it. Indeed, the locomotion activity of *Gryllus bimaculatus* was reported to be nocturnal in all studies by Tomioka and colleagues. Nevertheless, it was consistently diurnal in our setup. In accord with the report by Tomioka and Chiba (1982), we did anecdotally observe diurnal locomotion in nymphs. However, as noted, in our experiments diurnal locomotor behavior persisted in adults: in the anechoic chamber, LD crickets were foraging during the day and stridulating at night. The temperatures in our experimental chamber was 24-28 degrees Celsius, therefore, this could not be explained by rhythm reversal as a result of exposure to low temperatures as described in Ikeda and Tomioka (1993). The natural behavior of the species might differ between the laboratory colonies, and reflect some adopted colony-specific behavior. Additionally, there may be some indirect effects of the different motion detection methods used by both laboratories. Our method showed mostly diurnal locomotion, yet, we did observe in the LD control a peak in locomotion at sunset, preceding the beginning of stridulation behavior, which is in accord with the actograms shown in Tomioka and Chiba, 1982. All in all, we cannot state with certainty what causes this discrepancy between both laboratories. We agree with the referee that this issue is interesting and deserves careful attention, and have now added it to the discussion.

Revisions:

L260- L273: "In the present study we report on the temporal partitioning of field cricket behavior, with stridulation being predominantly nocturnal and locomotion predominantly diurnal. This is in contrast to previous studies in the same species that demonstrated the diurnal locomotion of nymphs, but nocturnal locomotor activity of adults [47]. The growth chamber and experimental set-up temperatures were 24-28oC. Hence, the above discrepancy cannot be explained by rhythm reversal as a result of

exposure to low temperatures [48]. The natural behavior of the species might differ among different laboratory colonies, and reflect certain adaptive colony-specific behaviors. Additionally, there may be some indirect effects of the different illumination and motion detection methods used in the different laboratories. While, as noted, our method revealed mostly diurnal locomotion, we did observe in the LD group a peak in locomotion at sunset, preceding the beginning of stridulation behavior, which is in accord with the actograms shown in [47] (Figure 1 therein). Although this has little impact on the main findings reported here, this issue deserves further investigation."

4. It may be better to describe how the authors determine the synchronization of the rhythms to given LDs, because the daily rhythms are often caused by masking effects of light that are bypassing the endogenous clock.

Response and revision: We thank the referee for pointing out this significant issue. The LD rhythms can, indeed, be a result of masking effects, caused by the diurnal illumination. We have now added a mention to this point, please see lines 313 – L314: "It should be noted that a period of 24h does not necessarily reflect a stable endogenous cycle but could also be the result of masking. "

5. L139: The authors stated that the rhythm was practically absent in LL treatment. This seems to be inadequate, because considerable number of animals showed rhythms under LL. Actually 17 and 24 animals were rhythmic under LL in stridulatory and locomotor activity, respectively. According to Table S1, these are about 70% and 60%. Then why do the profiles in Figure 3A, B show arrhythmic pattern?

Response: This is a very important point and we thank the referee for bringing this unclarity to our attention. The referee is of course correct in noting that a considerable number of animals showed rhythms under LL. These free-run rhythms could be determined for the single animal. However, at the population level, since the periods of the rhythms differed from one animal to the other, they were averaged. Figure 3A3 and 3B3 represent the overall mean rhythm of the population based on the mean of the individual rhythms. Following this comment, we rephrased the relevant section to clarify the difference of rhythmicity on the individual and the population level.

Revisions:

L185 – L186: "For both behaviors, averaging the free running data of the different LL individuals resulted in a practically absent rhythm."

6. L144-145: a significant period should be better.

Response and revision: Corrected. Thank you. The line now reads: L189 – L190: "Only individuals that exhibited a significant period (Table S1) were included in this analysis."

7. L145, 154: Why did you compare "medians" instead of means?

Response: In order to compare means using ANOVA, two assumptions have to be completed. The data have to be normally distributed, and a homogeneity of variances has to be achieved. Figure 4A,B shows significant differences among the treatment's variance, and hence, the assumption of the homogeneity of variances cannot be fulfilled. Therefore, we used non-parametric tests for this dataset, comparing the median instead of the mean. We thank the referee for the opportunity to re-review our statistical examinations.

Revisions:

L193 – L196: “Furthermore, the variance in the period of stridulation of all three ALAN treatments significantly differed from that in the LD treatment (Fig. 4A, Brown Forsythe test with Bonferroni correction).”

L199 – L201: “In addition, the variance in the period of locomotor behavior differed significantly between LL and both LD and LL2 treatments (Fig. 4B, Brown Forsythe test with Bonferroni correction).”

8. L146,148, 154, 155 and 156: Please provide SD for each condition. This is important to know variance.

Response and revisions: For the relevant analysis we used the median and not the mean. Therefore, the SD is not indicated, but the variance is presented in figure 4, where the black, horizontal lines the inter-quartal range 25% below and 25% above the median. We thank the referee for pointing out this unclarity and have added this indication of the variance to fig. 4 legend (please see our revision to comment 9).

9. Figure 3C, D: What do colored areas mean? Please explain what black vertical lines are.

Response: We thank the referee for noticing this absent information. The colored areas are Kernel histogram, rotated by 90 degrees. The black vertical lines represent the inter-quartal range 25% below and above the median. Therefore, the complete black line represents 50% of the data.

Revisions:

Figure legend 4: “Red lines represent medians, and individual males are plotted in colored dots. Colored areas are Kernel histogram, rotated by 90 degrees. Black vertical lines represent the inter-quartal range 25% below and 25% above the median.”

10. L175: Table S1 & Table S2. Table S2 is missing.

Response and revisions: Corrected and removed. A new table S2 was added to line 216.

11. L194: reference #67 is missing in the list.

Response and revision: Thank you, the reference was corrected and the sentence now reads: L287 - L288: ...”ALAN-exposed *Teleogryllus commodus*, in [51]”

12. L213: “locomotion activity revealed some resilience to ALAN”: Is there any possibility that the observed synchronized locomotor rhythm is a masking effect of light?

Response: Thank you for this very important comment. Indeed, locomotion behavior took place diurnally, while ALAN manipulations were conducted nocturnally. It is, therefore, very possible that the perceived resilience of locomotion rhythm periods reflects masking.

Revisions: L313 – L314: “It should be noted that a period of 24h does not necessarily reflect a stable endogenous cycle but could also be the result of masking.”

13. L214: Please insert comma after exposure.

Response and revision: Done. Thank you. L312: “ exposure,”

Appendix B

Reviewer comments

Response to editor: Accept with minor revisions

The authors have responded to most of the objections raised by the reviewers in the first round of peer review. They have clarified many of the methodological issues, performed the requested analysis and discussed the caveats to some of their results.

This study examines the effect of visible light pollution on two separate behaviours, locomotion and activity in the house cricket. It is well designed and has clear and convincing results showing how both locomotion and stridulation behaviour are altered in the presence of light pollution compared to a 12 L:12 D cycle. They compare this to an all light (12L:12L) treatment and find a progressively greater disruption in rhythmicity as they increase the intensity of the simulated light pollution in the different treatments, with the greatest affect visible in the all light condition. It is a useful contribution to the field and other than a few minor points, should be accepted for publication.

The abstract and introduction clearly identify the need and relevance for this research, the methodology targets the main question appropriately. The results are logically explained and the authors have made the figures clear and easy to understand. The discussion clearly addresses some of the potential shortcomings of the study.

Minor Issues

Abstract

Line 22: The term acrophase should be explained, even if in a few words in parenthesis.

Line 27-29: "The effects were mostly light-intensity-dependent, revealing an increase in the difference between the activity periods calculated for stridulation and locomotion in the same individual." Thesis statement is a vague, maybe use magnitude of disruption was light intensity dependent?

Introduction

Line 69: "is of course crucial for reproduction", avoid using implied obviousness of the fact since it may not be obvious to readers not familiar with the system.

Line 73: Although this has been mentioned in the discussion, it might be worth mentioning here what the expected patterns of activity in the control condition, based on our current understanding about their natural history. Is stridulation behaviour more common at night and motion during the day. Maybe just a few sentences about the variation in known diel-periods for locomotion and stridulation in crickets.

Materials and Methods

Line 82: What was the range of humidity they were at? It would be useful for future studies to know the exact conditions.

Line 117: Can you explain the use of a variable threshold (2-10%)?

Line 115: What does IR represent here, shouldn't this refer to the name and manufacturer of the camera? Does the camera itself produce the IR source, this needs to be elaborated. Also is it on all the time, or does it come on only when the light levels are low.

Results

Line 158: Preferably add a subheading for body size relations.

Discussion

It would be useful to justify the choice of some of the statistical analysis used in the paper, the use of median in some cases and mean in others as well as the motivation for the phase analysis if the paper does not exceed the word limit.

Figures

Why is the naming convention across subfigures not consistent? Is this the journal convention, or is there a justifiable need to use A1, A2, etc. for example to distinguish stridulation and locomotion behaviour?

Figure 2: It would be better to keep the night and day colours consistent in the same panel. A and B have different shades of yellow. Just a suggestion, B1 and B2 seem to take a disproportionate amount of space in the figure. B1 and B2 could easily be included in A1 and A2 as small plots, but this is an aesthetic concern and does not affect the readability of the graph.

Figure 3-5 are very well presented and the patterns are very obvious even on visual inspection.

Appendix C

29th August 2021

Resubmission: manuscript ID RSPB-2021-1626

Dear Editor,

Thank you very much for your notice of acceptance of our manuscript. We greatly appreciate the contribution of the editor and reviewers to this paper. We have made some further minor changes and corrections to the paper following the final comments, as detailed below.

Best Regards,

Amir Ayali

e-mail: ayali@post.tau.ac.il

Reviewer(s)'	Comments	to	Author:
--------------	----------	----	---------

Referee 3:

1. The authors should describe how they calculated diurnal and nocturnal activity in LL animals in Materials and Methods section.

The following lines were added to the *Data processing and statistical analysis subsection* in the Materials and Methods, L139 – L140: “The analysis of the nocturnal and diurnal activity in the constant daylight (LL) crickets was based on the objective day and night periods.”

2. L172: I think LL5 should read LL2.

The text as it appears is correct: L177 – L180: “the median of the normalized diurnal, as well as nocturnal LD stridulation activity level significantly differed from the LL5 and LL treatments ($p < 0.007$ for all, Kruskal-Wallis test with Dunn’s multiple comparisons, Fig 3A1, Table S1).

Referee 4:

Abstract

1. Line 22: The term acrophase should be explained, even if in a few words in parenthesis.

A short definition was added the Material and Methods section, L153: “The mean acrophase (the time at which the peak of a rhythm occurs)”.

2. Line 27-29: “The effects were mostly light-intensity-dependent, revealing an increase in the difference between the activity periods calculated for stridulation and locomotion in the same individual.” Thesis statement is a vague, maybe use magnitude of disruption was light intensity dependent?

L26 – L27: Changed to “The magnitude of disruption was light intensity dependent”.

Introduction

3. Line 69: “is of course crucial for reproduction”, avoid using implied obviousness of the fact since it may not be obvious to readers not familiar with the system.

L69 – L70: “Of course” was omitted and now read: “Stridulation, explicitly calling songs used for communication and female attraction, is crucial for the reproduction of the species.”

4. Line 73: Although this has been mentioned in the discussion, it might be worth mentioning here what the expected patterns of activity in the control condition, based on our current understanding about their natural history. Is stridulation behaviour more common at night and motion during the day. Maybe just a few sentences about the variation in known diel periods for locomotion and stridulation in crickets.

L60 – L63: “Crickets (Gryllidae) have been widely utilized as models for the study of insect physiology, neurobiology, and behavior, including circadian activity [29,30]. They are known to demonstrate clear diel cycles in two fundamental behaviors, stridulation and locomotion [31-35].”

L74 - L77: This paragraph was changed to read: “Here we studied the effects of exposing male *G. bimaculatus* crickets to lifelong ALAN on their stridulation and locomotion patterns. This is one of very few examples in which the two behaviors have been monitored simultaneously [32,34,35], conducing to a more comprehensive understanding of the ALAN-induced behavioral effects.”

Materials and Methods

5. Line 82: What was the range of humidity they were at? It would be useful for future studies to know the exact conditions.

We have not consistently monitored the humidity in our experimental chambers. When occasionally tested humidity was between 60-70% (as now noted, please see L91).

6. Line 117: Can you explain the use of a variable threshold (2-10%)?

L116 – L117: We are grateful for the referee for drawing our attention to this misprint. The sentence was changed to read: “A threshold was defined as a change of more than 2% of pixels in the picture.”

7. Line 115: What does IR represent here, shouldn't this refer to the name and manufacturer of the camera? Does the camera itself produce the IR source, this needs to be elaborated. Also is it on all the time, or does it come on only when the light levels are low.

L115: IR appears first in line 114 where it is spelled out as infra-red. We appreciate this comment and have rephrased the sentence to be clearer. IR is constantly emitted by the camera itself (manufacture unknow). L113 – L116: “Locomotion activity was captured from above at 2 frames/sec, by an infra-red (IR) surveillance camera (constantly emitting IR light, peak: 799 nm), connected to a computer using the Active WebCam program (PY Software) for motion detection [39]” It should be kept in mind that the insects do not see IR.

Results

8. Line 158: Preferably add a subheading for body size relations.

L162: Added “ Crickets’ morphological characteristics”

Discussion

9. It would be useful to justify the choice of some of the statistical analysis used in the paper, the use of median in some cases and mean in others as well as the motivation for the phase analysis if the paper does not exceed the word limit.

Some words of explanation regarding the utilized statistical analysis were added to the Data processing and statistical analysis subsection under the Materials and Methods.

L149 – L152: “ It should be noted that most of our data is characterized by inequality of variance and/or by non-normal distribution. Hence, medians rather than means were often presented, and non-parametric statistical tests were utilized. ”

Figures

10. Why is the naming convention across subfigures not consistent? Is this the journal convention, or is there a justifiable need to use A1, A2, etc. for example to distinguish stridulation and locomotion behaviour?
11. Figure 2: It would be better to keep the night and day colours consistent in the same panel. A and B have different shades of yellow. Just a suggestion, B1 and B2 seem to take a disproportionate amount of space in the figure. B1 and B2 could easily be included in A1 and A2 as small plots, but this is an aesthetic concern and does not affect the readability of the graph.

We went carefully over all figures to make sure that they are consistent in our use of color shades and subheadings. Additionally, we have changed the subheading in figure 2 and accordingly the figure legend and resized all subheadings, to be more proportionate. We thank the referee for these constructive, aesthetic suggestions.